# Changes to information in working memory depend on distinct removal operations

Hyojeong Kim[1], Harry R. Smolker[2], Louisa L. Smith[2], Marie T. Banich[2] & Jarrod A. Lewis-Peacock [1✉]

Holding information in working memory is essential for cognition, but removing unwanted thoughts is equally important. Here we use multivariate pattern analyses of brain activity to demonstrate the successful manipulation and removal of information from working memory using different strategies including suppressing a specific thought, replacing a thought with a different one, and clearing the mind of all thought. These strategies are supported by distinct brain regions and have differential consequences for allowing new information to be encoded.

[1] Department of Psychology, University of Texas at Austin, Austin, TX 78701, USA. [2] Institute of Cognitive Science, University of Colorado, Boulder, Boulder, CO 80309, USA. ✉email: jalewpea@utexas.edu

The ability to actively hold information in mind, known as working memory (WM), is a central component of cognition necessary for guiding adaptive behavior. Due to the limited capacity of WM[1–3], the ability to remove irrelevant information from mind is equally essential[4], and deficits in this ability characterize many psychiatric disorders including depression, generalized anxiety disorder, post-traumatic stress disorder (PTSD), and obsessive compulsive disorder[5–7]. Understanding the mechanisms by which people remove thoughts from mind has been challenging, as it is difficult if not impossible to confirm through participant self-report or indirect behavioral measures that a thought has indeed been expunged in the brain. Recent proposals suggest that the capacity limitations of WM can be dealt with by either taking an item out of the focus of attention within WM or by removing it entirely from mind[8–10]. This latter process is proposed to free up capacity in WM which facilitates the encoding of new information[4].

Previous studies have proposed multiple ways that information can be removed from WM, including passive decay[11,12], interference[13,14], and the engagement of cognitive control strategies[15,16]. To accomplish motivated forgetting of long-term memories, inhibitory control processes in prefrontal cortex can suppress awareness of unwanted memories either during encoding or retrieval[17]. In related work, our previous study[15] focused on three distinct strategies for removing a thought from mind: to replace that thought with another thought, to suppress that specific thought, and to clear the mind of all thoughts. A mass-univariate analyses of functional magnetic resonance imaging (fMRI) data revealed that a hierarchy of brain regions involved in cognitive control, including parietal, dorsolateral prefrontal, and frontopolar regions, were engaged to varying degrees depending on the way information was removed from WM. However, it is not known how these distinct operations impact the neural representations of the information being removed, nor what their impacts are on subsequent encoding in WM.

Here we were motivated by recent advances in the marriage of machine learning and neuroimaging[18–20] to investigate the neural consequences of distinct removal operations in WM. We recorded fMRI data while participants encoded images from three stimulus categories (faces, fruit, and scenes) into WM and then performed cognitive operations on them (maintain, replace, suppress, and clear) (Fig. 1). Using machine learning approaches we analyzed these data with three primary objectives: (1) to demonstrate a differentiation between multivariate neural patterns associated with each method of removal; (2) to characterize how well removal operations limit access to the representation of the removed item; and (3) to quantify the degree to which the removal operations reduce the strength of the removed item so as to facilitate encoding of subsequent items.

To anticipate, we provide evidence that indeed information can actively be removed from WM and that there are at least three distinct ways to do so. Furthermore, we show that some of these ways (replacing and clearing) appear to act by taking an item out of the focus of attention, which deactivates its neural representation but leaves the information intact, while another way (suppressing) acts by expunging an item thus freeing up WM capacity for encoding other information.

## Results

**Neural dissociation of removal operations.** With regards to our first objective, which was to determine whether indeed the four operations of interest—maintain, replace, suppress, and clear—are distinct, we trained fMRI pattern classifiers on whole-brain data from each trial of the central study to determine if patterns of brain activation could differentiate the four different cognitive operations (Fig. 2a). As expected, all operations were reliably classified (area under the ROC (AUC): averaged across operations, $M = 0.74$, SEM $= 0.016$; one-sample $T$-test: all effects were more reliable than $T(49) = 12.25$, $P = 1.11e{-}16$, $d = 1.733$, 95% CI [0.17, 0.24], see Supplementary Table 1 for the full statistics), and distinguishable from one another, in line with our hypothesis that these operations indeed have dissociable neural origins. Importantly, although both suppress and clear require that nothing be held in mind, they were highly differentiable from each other (classifier accuracy, $T(99) = 12.46$, $P < 0.001$, $d = 1.246$, 95% CI [0.25, 0.34]). Moreover, both were differentiable from replace ($T(149) = 22.42$, $P < 0.001$, $d = 1.831$, 95% CI [0.33, 0.45]), suggesting that participants were indeed engaging in different strategies to implement these different removal instructions. The classifier importance maps[21] for these operations shown in Fig. 2b indicate that unique regions distributed across frontal, parietal, and occipital regions were critical for identifying the engagement of each operation. These regions are consistent with those identified via a prior univariate fMRI analysis[15], and the pattern of which was replicated in a univariate fMRI analysis of the current data set (see Supplementary Fig. 1). We replicated these within-subject classification results using between-subject classifiers (accuracy $M = 0.4$, SEM $= 0.012$, all effects are more reliable than $T(48) = 6.4$, $P = 6.2e{-}08$, $d = 0.914$, 95% CI [0.06, 0.12], Fig. 2a, right, Supplementary Table 1) in which all individuals' voxels in MNI space were aligned anatomically. The success of this procedure, combined with the group-level univariate results and classifier importance maps, demonstrates that similar neural processes were recruited for each operation across participants.

**Changes to information being removed.** With regards to our second objective, which was to determine the degree to which each of these removal operations limits access to the representation of information in WM, we also utilized machine-learning classifiers. In this case, however, the classifiers were trained not on the operation performed but rather with regards to the nature of the information in WM, that is, the (visual) categories from which our items were drawn. To create these classifiers, participants performed a separate functional localizer task in the MRI scanner prior to performing the central study. Participants were presented with the same items used in the central study, one at a time. To promote attention towards the stimuli, they rated the desirability of each item on a four-point scale (see the "Methods" section for details). Importantly, images belonged to three categories (faces, fruits, and scenes) with three subcategories within each category (face: actor/politician/musician; fruit: apple/grape/pear; scene: beach/mountain/bridge). Brain activity patterns evoked by these images were successfully differentiated by applying multivariate pattern analyses at the category-, subcategory-, and item-specific levels (Fig. 3, see the "Methods" section and Supplementary Table 2 for details), and therefore these data could be used to evaluate changes in the representation of items within WM during the central study.

Once the pattern of activity for each category had been identified in the localizer data, these patterns could then be applied to the central study data. Prior work has demonstrated that using neural pattern classifiers to decode the status of information in WM reveals only the subset of information in WM that is currently in the focus of attention[8,10]. Unattended information in WM may not exert a sustained neural signature, but this information is nonetheless in WM as it can be identified using external stimulation methods or by redirecting internal attention to that information[22,23]. Therefore, the results for decoding representational status that are described next are best

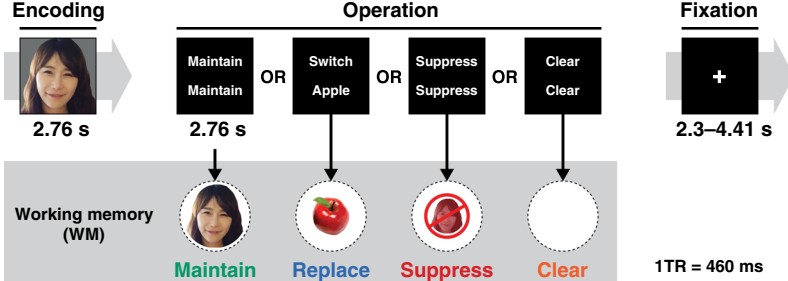

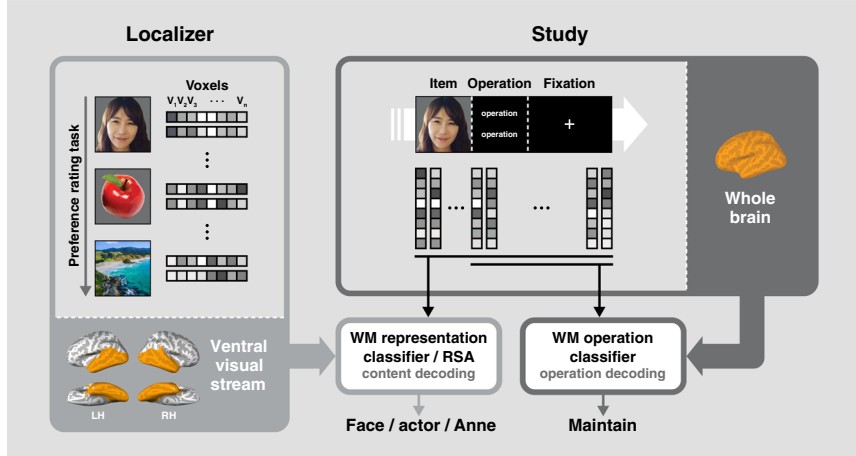

**Fig. 1 Behavioral paradigm and fMRI procedures. a** In the central study, on each trial, participants viewed a picture (a face, fruit, or scene) for 2.76 s. While images of famous faces (actors, musicians, and politicians) were used in the experiment, the authors' faces are featured in the figures instead. Next, a screen appeared for 2.76 s indicating which one of four different cognitive operations (maintain, replace, suppress, clear) should be applied to the just-viewed image. This screen was followed by a screen with a fixation cross whose duration varied from 2.3 to 4.41 s. **b** Brain data from a perceptual localizer task (left) were used for multi-voxel pattern analysis (MVPA) for stimulus category-level and subcategory-level decoding, and item-level representational similarity analyses (RSA) of voxel activity in the ventral visual stream (VVS) ROI. These analyses were then applied to brain data from the central study to enable decoding of information within the focus of attention in working memory (WM). Note that whole brain data was used for the subcategory classifier to capture semantic differences across subcategories (e.g., actors vs. musicians). Additionally, whole brain data from the central study was used for pattern classification of the cognitive operation being performed on each trial.

interpreted as reflecting the attentional state of an item in WM, rather than the presence or absence of that item in WM. The latter issue is addressed by additional analyses evaluating consequences of various methods of removal on subsequent encoding of new items.

As would be expected if participants could differentially manipulate items, when participants were instructed to replace an item, that item's neural category representation dropped to baseline more quickly than when they were instructed to maintain the item (windows 1–5, more reliable than $T(49) = 3.55$, $P = 8.50e−04$ (FDR corrected), $d = 0.503$, 95% CI [0.003, 0.03], Fig. 4a, Supplementary Table 3). Furthermore, an increase in classifier evidence was observed for the category of the new item to which individuals switched their attention on replace trials. To statistically evaluate this increase, we compared it with an empirical baseline of classifier evidence for an irrelevant category from suppress and clear trials. This baseline was computed separately for each participant by randomly sampling an irrelevant stimulus category from each suppress and clear trial (there were no differences between these trial types for all time windows, less reliable than $T(49) = 1.80$, $P = 0.390$, $d = 0.255$, 95% CI [−0.01, 0.04], and thus suppress and clear trials were averaged for the baseline). For example, if Anne Hathaway (a

face) was suppressed, we would select randomly the fruit or scene category evidence from that trial to contribute to the baseline. On replace trials, the increase in classifier evidence for the replacement item rose significantly above this baseline (windows 4 and 5, more reliable than $T(49) = 6.73$, $P = 4.39e−08$, $d = 0.951$, 95% CI [0.03, 0.09], Fig. 4a, Supplementary Table 3). This verifies that evidence of replacement of one item with another was found only for *replace* trials, and not for either suppress or clear trials.

Additionally, the three removal operations (replace, clear, suppress) had unique impacts over time on the representation of the item being removed. These differences are highlighted by depicting the time course of removal of information for each operation relative to the maintain condition (Fig. 4b, top). Qualitatively, the representation decoding trajectory for clear trials falls in between replace and maintain. Somewhat para-doxically the trajectory for suppress is quite similar to that for maintain, suggesting that a representation must be within the focus of attention in order for it to be suppressed[24].

Note that we decoded the item-specific neural patterns being represented rather than neural activation intensity per se. Interestingly, the univariate neural activation in the ventral stream was decreased for suppress compared to maintain, while

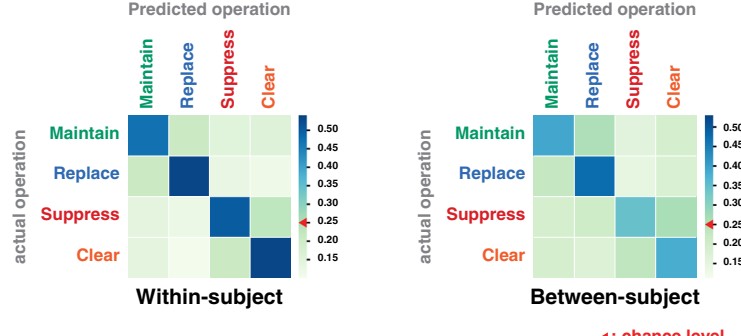

**a** WM operation classifier confusion matrix
from study: 6-fold (run) cross-validation

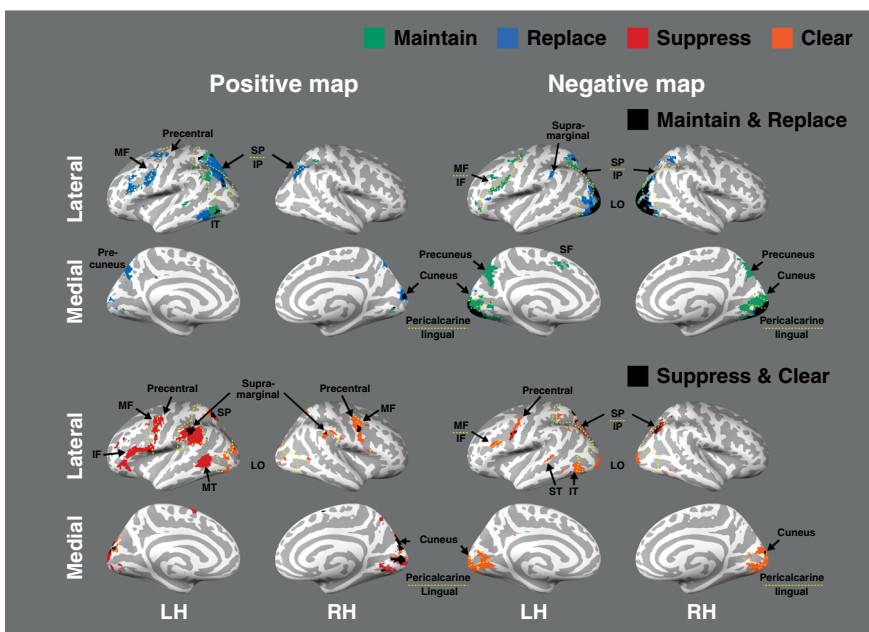

**b** WM operation classifier importance maps

**Fig. 2 WM operation multivariate pattern analysis. a** Classifier confusion matrix for decoding the operation being engaged on each trial of the central study with within-subject classifiers [$N = 50$ participants; left] and between-subject classifiers [$N = 49$ participants; right]. Source data are provided as a Source Data file. **b** The classifier importance maps highlight voxels that were identified as important for identifying each operation (positive: increased activity is important; negative: decreased activity is important). The cortical labels were adapted from Mindboggle-101 provided in FreeSurfer[58]. S/M/I superior/middle/inferior, F/P/T frontal/parietal/temporal, LO lateral occipital, LH/RH left/right hemisphere. The different operations are colored according to the legend in **a**, with black representing overlap between maintain & replace (top) and suppress & clear (bottom). See Supplementary Table 1 for the statistics.

the multivariate results were equivalent for these two operations. This pattern suggests that suppression may promote sharpening of the representation to selectively suppress the target. Repeated presentations of a stimulus, which produces a reduction of activation as assessed by univariate approaches, has been shown to be associated with either increased multivoxel pattern classifier evidence (i.e., representational sharpening), or with decreased evidence[25,26]. Critically, our results propose that suppression may actively identify and target the representation of the item in WM that is to be removed rather than simply inhibiting WM activity in general. This finding is consistent with our recent study demonstrating that intentional forgetting of a picture stimulus produced stronger (sharper) multivariate representations of the targeted item during the forgetting attempt[24].

Data were evaluated for five 1.38 s (3 TR) time windows beginning at the onset of the manipulation instruction at 2.76 s after stimulus onset and extending to 9.66 s post-stimulus onset.

Results show that replace had the largest drop in category classifier evidence for the WM item, followed by clear and then suppress. Pairwise comparisons indicate that replace was reliably lower than clear (windows 2–5, more reliable than $T(49) = 3.38$, $P = 0.002$, $d = 0.479$, 95% CI [0.01, 0.03]) and suppress (all 5 windows, more reliable than $T(49) = 3.8$, $P = 3.95e-04$, $d = 0.538$, 95% CI [0.01, 0.03]). Differences between *clear* and suppress only emerged towards the end of the trial (windows 4 and 5, more reliable than $T(49) = 2.64$, $P = 0.011$, $d = 0.373$, 95% CI [0.003, 0.03]). These analyses were repeated using item-level decoding (Fig. 4b, bottom) which revealed a similar ordering of results for the degree of removal (replace > clear > suppress; all pairwise are more reliable than $T(49) = 3.71$, $P = 5.35e-04$, $d = 0.524$, 95% CI [0.02, 0.05] for significant time windows, see Supplementary Table 3 for the full statistics).

In addition to the degree to which classifier evidence drops, we can also examine when that drop becomes significant. In Fig. 4b,

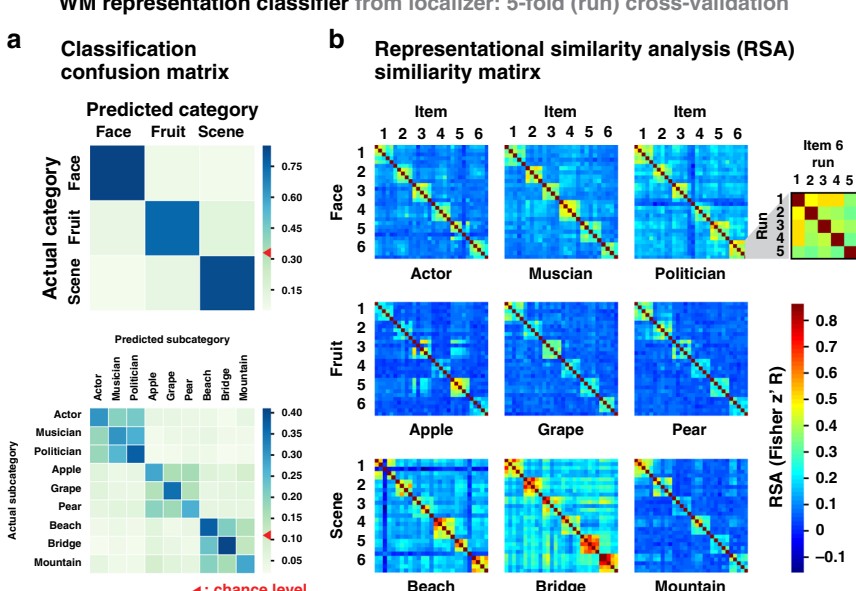

**Fig. 3 WM representation multivariate pattern analysis. a** Classifier confusion matrices for category-level classification (in the ventral visual stream, VVS) and subcategory-level classification (whole brain), **b** and item-level RSA (VVS) on the localizer data for $N = 50$ participants. Within each subcategory there were six items (e.g., six actors) each of which was shown once in each of five runs. The inset provides an expanded view of the similarity matrix of the item-level RSA across the five runs for a single item (item 6 from the politician subcategory). See Supplementary Table 2 for the statistics. Source data are provided as a Source Data file.

significance markers (triangles) indicate the initial time window in which the classifier evidence for the memory target in one of the removal conditions is statistically below the classifier evidence from maintain. At the category-level (top row), replace shows the earliest onset of removal (window 1: 2.76–4.14 s; one-sample $T$-test, $T(49) = 3.55$, $P = 8.50e{-}04$, $d = 0.503$, 95% CI [0.01, 0.03]), followed by clear (window 2: 4.14–5.52 s; $T(49) = 2.96$, $P = 0.006$, $d = 0.419$, 95% CI [0.003, 0.02]) and then suppress (window 4: 6.9–8.28 s; $T(49) = 3.97$, $P = 0.001$, $d = 0.562$, 95% CI [0.01, 0.03]). Complementary item-level decoding analyses were then performed on these data using a weighted representational similarity analysis (RSA; see the "Methods" section). Estimates of the onset for removal of information from the focus of attention differed for the item-level analysis (bottom row), in which suppress showed the earliest indication of removal (window 2: 4.14–5.52 s; $T(49) = 2.97$, $P = 0.008$, $d = 0.42$, 95% CI [0.01, 0.03]). The discrepancies between the item-level vs. category-level analyses regarding onset of removal from attentional focus might be due to reduced sensitivity in these two types of analyses. However, they might also reflect a meaningful cognitive difference such that suppression first impacts item-level details of a stimulus before impacting its general category-level information. Further research is needed to differentiate between these two possibilities.

Taken together, these brain decoding analyses demonstrate that there are at least three unique cognitive operations that can be invoked to remove an item from WM. By tracking the attentional focus of WM during the removal attempts, we found that replacing an item had the largest impact on that item's attentional status, reducing it at both the category and item levels. Clearing the mind of all thought yielded similar results to that of replace but to a lesser degree. The difference between these two conditions may arise because in replace, a new representation overrides the prior representation in the focus of attention, or because in the clear condition attentional focus is more diffusely removed as to apply to all thought. Finally, suppress had the least impact on the item's attentional status, having minimal effects at

the category level, while subtly reducing the degree to which the stimulus representation remains in the focus of attention at the item level. This result suggests that, similar to recent findings in directed forgetting, it may be necessary to maintain at least some information about an item in order to target it for suppression[24].

**Impacts on encoding after removal.** With regards to our third objective, we evaluated the degree to which these operations show distinct consequences on encoding of subsequent items into WM. As discussed above, the WM representation decoding results in Fig. 4 reflect the attentional state of the items in WM, not their mnemonic state. It is possible for a WM item to become removed from the focus of attention, thus un-decodable from brain activity, but not forgotten and easily retrieved back into the focus of attention[4]. Therefore, to address the question of whether an item has truly been removed from WM, and not just neurally deactivated, we investigated the impact of these different removal operations on subsequent encoding of new information, which should reflect the status of the information that was to be removed. The logic here is that if an item has indeed been removed from WM, it should not interfere with the encoding of new information. However, if the item was not removed from WM, it should produce proactive interference during the subsequent encoding of new information. Proactive interference[27] occurs when previously learned information interferes with new learning, leading to longer response times (RTs) and more false alarms[28,29]. The ability to encode multiple items into WM at the same time is limited by the extent to which those items are represented by separate neural populations[30]. Based on the idea that similar items in semantic memory involve somewhat overlapping representations[31,32], proactive interference should limit the ability to clearly encode a subsequently presented item that is semantically similar. Thus, to the degree that an item is indeed removed from WM, it should facilitate encoding of subsequent items by reducing proactive interference[33]. For example, effectively removing an image of Anne Hathaway from WM should impact the subsequent encoding of Bernie Sanders more than

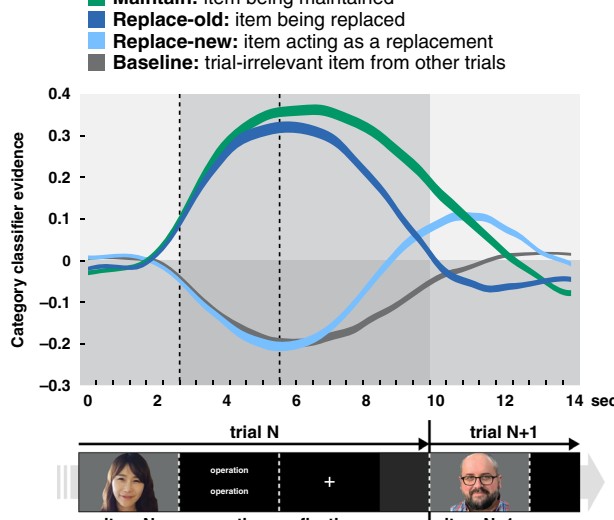

**a** Timecourse for neural decoding of a WM item

**Maintain:** item being maintained
**Replace-old:** item being replaced
**Replace-new:** item acting as a replacement
**Baseline:** trial-irrelevant item from other trials

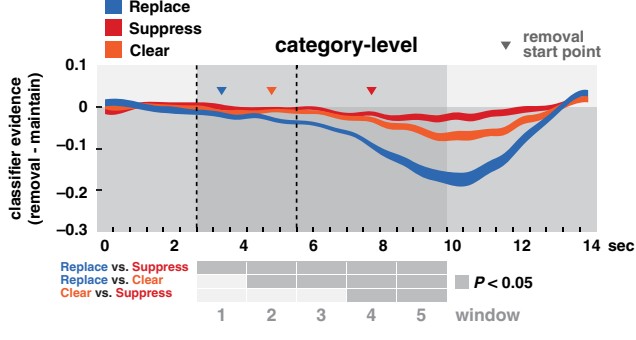

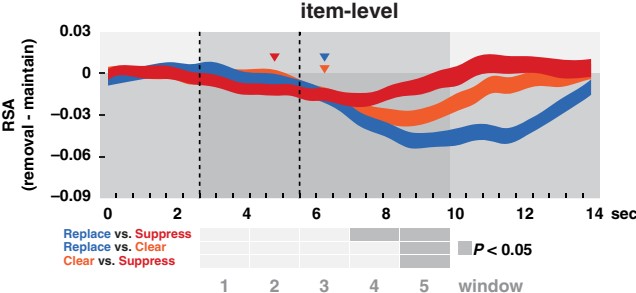

**b** Trajectory for removal of an item from WM

**Fig. 4 Neural decoding of WM representations. a** Group-averaged category-level fMRI pattern classifier evidence scores are shown for 14 s after the onset of each trial, shown separately for the item on maintain trials (green), the item being replaced on replace trials (blue), the item that serves as a replacement on replace trials (light blue), and an empirical baseline of trial-irrelevant items selected randomly from suppress and clear trials (gray). Data between discrete data points at each TR (460 ms) were interpolated and are presented as ribbons whose width represent the mean ± 1 SEM for $n = 50$ participants. Triangle indicates the replace-new start point defined as the first 3-TR window (1.38 s) in which the decoding evidence for the replace-new item was reliably greater than baseline. **b** Category-level (top) and item-level (bottom) decoding evidence for each of the three removal operations, replace (blue), suppress (red), and clear (orange), subtracted from that for maintain trials at each timepoint. Deviations below zero indicate a reduction in information in brain activity below maintenance of the to-be-removed item. Triangles indicate the removal start point for each condition, defined as the first 3-TR window (1.38 s) in which the decoding evidence for one of the trial types was reliably below zero (start point for classifier evidence: replace, $P = 8.5e-04$; clear, $P = 8.73e-04$; suppress, $P = 0.001$; for RSA: suppress, $P = 0.008$; replace, $P = 0.002$; clear, $P = 0.002$). Width of the data ribbons indicates the mean ± 1 SEM for $n = 50$ participants. Statistical comparisons between conditions are indicated in the charts below each plot with dark gray cells indicating statistical significance, *$P < 0.05$, two-sided, repeated measures pair-wise $T$-tests (FDR corrected). For the category-level trajectories of classifier evidence, the statistical comparisons were $P$s < $3.95e-04$ for replace vs. suppress across the significant windows, $P$s < $0.002$ for replace vs. clear, and $P$s < $0.028$ for clear vs. suppress. In the item-level trajectories of RSA correlations, the statistical comparisons were $P$s < $4e-04$ for replace vs. suppress, $P = 4.41e-04$ for replace vs. clear, and $P = 5.35e-04$ for clear vs. suppress. See Supplementary Table 3 for full statistics. Source data are provided as a Source Data file.

subsequent encoding of an apple because the representations of Anne Hathaway and Bernie Sanders have more overlapping features and neural representations than do Anne Hathaway and the apple[30].

We examined the impact of the different operations on subsequent encoding via the brain decoding results. Our approach was to use item-level decoding via RSA to compute the encoding fidelity for each image that was viewed in the central study. This measure reflects the correspondence between the average response pattern to a given image during the functional localizer task and the neural response pattern for an image viewed during the central study (Fig. 5a, see the "Methods" section). This measure was then examined for the image presented in each trial based on two features of the previous trial: the category of the previous image (same or different) and the operation performed on that previous image (Fig. 5b). To the degree that the representation of the item in the prior trial lingers in WM, it

should reduce the encoding fidelity of related items on the subsequent trial (i.e., those in the same category) and produce proactive interference (i.e., worse encoding fidelity for same category vs. different category images). However, if it has indeed been removed from WM, such proactive interference should be reduced, and potentially even reversed[34].

Providing evidence that such proactive interference can indeed occur, following maintain trials, there was a reduction in encoding fidelity for new items that were of the same category as the item that was maintained relative to items from a different category ($T(49) = 2.96$, $P = 0.005$, $d = 0.418$, 95% CI [0.01, 0.05]; Fig. 5b). This result confirms that holding an item in mind without deliberately expunging it leads to proactive interference when encoding new information into WM. Both replace and clear also produced proactive interference effects (more reliable than $T(49) = 2.35$, $P = 0.023$, $d = 0.332$, 95% CI [0.003, 0.05], Supplementary Table 4) of similar magnitude, suggesting that neither of these mechanisms removed the information of the initial item from WM. Note that results for replace here are with respect to the first item on those trials, reflecting only those trials in which the next item was from a different category than the replacement item from that trial (e.g., replace trial $N$: Category 1 ⇒ switch to Category 2; trial $N + 1$: {Category 1 (for same) or Category 3 (for different category)}). Results for replace trials with respect to the replacement item also show evidence of proactive interference on the next trial (see the "Methods" section). Critically, for suppress trials this interference was eliminated, suggesting that the suppress operation is effective in removing or reducing information in WM. When comparing across operations, the same-minus-different encoding fidelity was significantly higher for suppress than the other operations (one-way ANOVA, $F(3, 147) = 11.76$, $P = 6.00e-07$, $\eta^2 = 0.193$;[35] pairwise

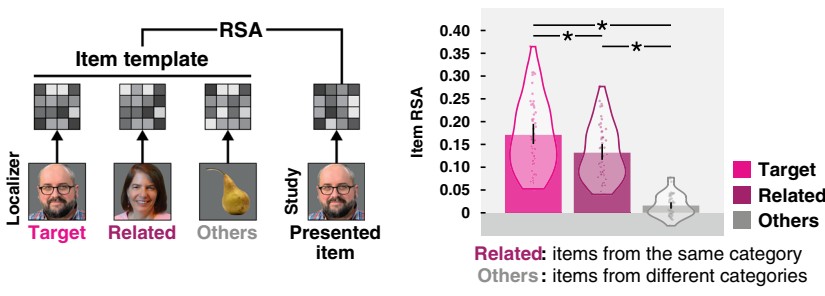

**a**  **Encoding fidelity: item-level RSA decoding**

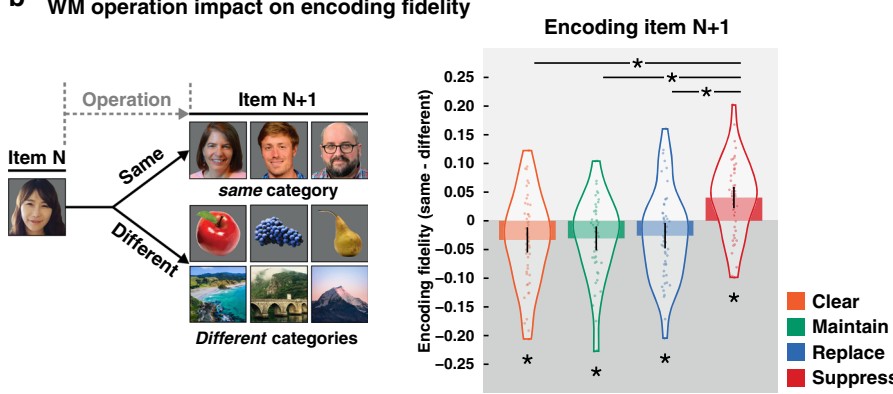

**b**  **WM operation impact on encoding fidelity**

**Fig. 5 Impact of cognitive operations in WM on fidelity of subsequent encoding. a** (Left) Item-specific patterns of brain activity were used as templates for comparison with each item presented during the central study. (right) The RSA data confirm that this analysis was sensitive enough to identify the presented (target) item on each trial. There was a significantly higher similarity when between the target template from the localizer task and the presented item in the central study, than with either items from the same category (related) or items from different categories (others). *$P < 0.05$, two-sided, repeated measures pair-wise $T$-tests (Tukey–Kramer corrected). Across pairwise comparisons, $P$s $< 1.11e{-}16$. **b** Each image was sorted based on the preceding item and operation. Bar graphs on the right reflect the encoding fidelity on the $N + 1$ trial of same-category images vs. different-category images (used as a baseline) following each operation type. Values below zero indicate proactive interference for encoding same-category images. The only operation that did not lead to proactive interference was suppress, suggesting that only it is effective in expunging the item on the prior ($N$) trial from working memory. *$P < 0.05$, two-sided, repeated measures pair-wise $T$-tests (Tukey–Kramer corrected). Across the significant comparisons, $P$s $< 0.001$. Violin plots show the full distribution of $n = 50$ participants with dots corresponding to individual participants. Error bars indicate 95% CI. See Supplementary Table 4 for full statistics. Source data are provided as a Source Data file.

comparisons with Tukey–Kramer correction, more reliable than $T(49) = 4.73$, $P = 1.11e{-}04$, $d = 0.669$, 95% CI [0.04, 0.09]) with no differences across the other operations (less reliable than $P = 0.968$, $d = 0.065$, 95% CI [−0.03, 0.04]). Note that there were no reliable differences for the different-category encoding fidelity across operations indicating an equivalent baseline (less reliable than $P = 0.096$, $d = 0.335$, 95% CI [0.003, 0.03]), with differences only observed for the same-category analysis (more reliable than $P = 7.22e{-}04$, $d = 0.588$, 95% CI [0.02, 0.06]). This pattern of results is consistent with prior work showing that engaging memory suppression through directed forgetting can reduce proactive interference in WM, an effect believed to result from the attenuation of the representation of the to-be-forgotten information[33]. In fact, suppression here led to a relative advantage for encoding information from the same-category as compared to a different-category on the next trial ($T(49) = 4.41$, $P = 5.66e{-}05$, $d = 0.624$, 95% CI [0.02, 0.06]). While speculative, this proactive facilitation may result from lingering categorical information in WM that has been stripped of item-specific details from the suppress operation. This generic category information may then be more easily repurposed to encode the new same-category stimulus with high fidelity.

These encoding fidelity results might be considered surprising based on the WM representation decoding results from Fig. 4b,

which show that replace and clear led to the greatest reduction in information corresponding to the item being removed from WM. Yet, these operations did not reduce proactive interference on the subsequent trial. It was only the suppress operation that reduced interference, and in fact, facilitated subsequent encoding for related information.

While this pattern may seem paradoxical, it likely indicates that suppression is a process which modifies the active representation of an item in some way that successfully reclaims the neural resources used to keep the representation active in WM[24]. Together, these results suggest that for an item to be truly removed from WM, that mere deactivation of its neural representation (and hence downgrading of its attentional status) is insufficient. This dovetails with recent ideas about a temporary removal process[4] by which an item can be removed from the focus of attention in WM but remains dormant so as to be easily accessed and reactivated. This process reflects only the temporary inattention to an item, rather than its removal per se. Instead, an additional process—here invoked by an instruction to suppress that item—is required to reduce its representation in WM and eliminate its interference on future processing. This finding is in line with the idea of a permanent removal process involving the active unbinding of an item from its context in WM that truly eliminates the information from mind[4,36].

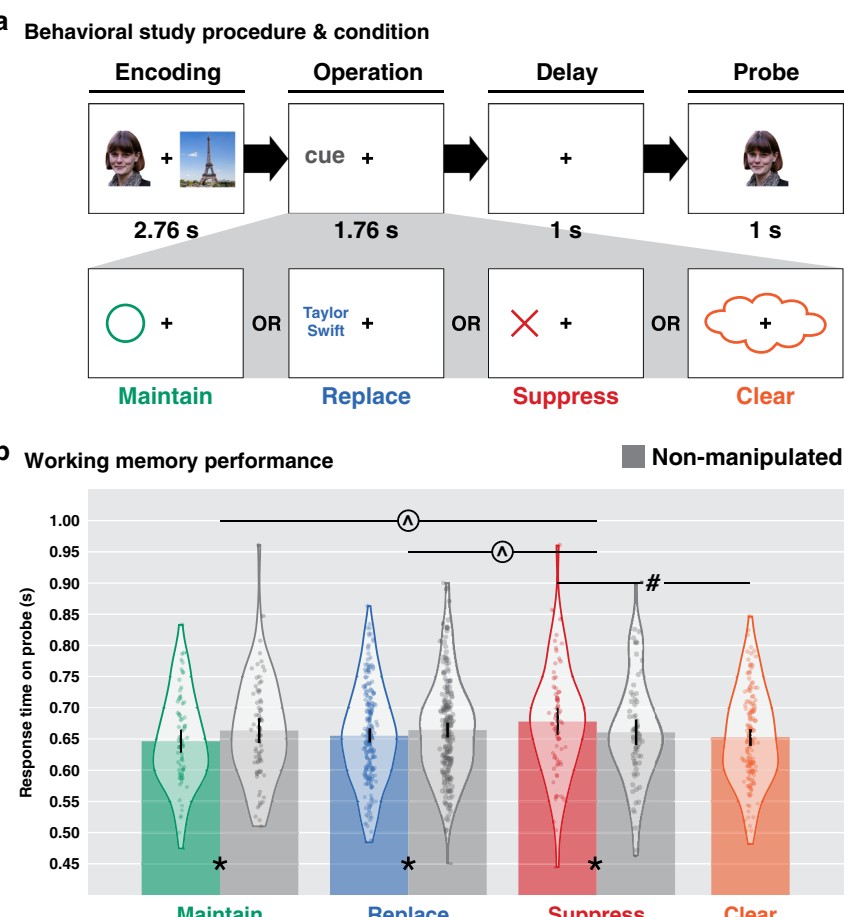

**Fig. 6 Behavioral study procedure and results. a** Each trial started with two items (one face and one scene) for 2.76 s, followed by a cue for 1.76 s indicating which cognitive operation should be applied to the item that had appeared at the cued location (i.e., the manipulated item). Each cue was linked to one of the four operations based on its shape and color. Then, a 1-s retention interval appeared with central fixation until a single WM probe appeared. The participants were instructed to respond whether they saw the probe item at the beginning of the trial (regardless of whether or not an operation was performed on the item) as fast as possible within 1 s. **b** Response times (RTs) for correct Yes probes for different operations by probe type (manipulated, non-manipulated) of the WM items. Data for manipulated items are color-coded by operation and data for non-manipulated items are shown in gray across all operations. Violin plots show the full distribution of $n = 208$ participants with dots corresponding to individual participants. Error bars indicate 95% CI. *Paired $t$-test, two-sided, uncorrected, Maintained vs. Non-Maintained $P = 0.016$, Replaced vs. Non-Replaced $P = 0.014$, Suppressed vs. Non-Suppressed $P = 0.025$; # independent $t$-test, two-sided, uncorrected, Suppressed vs. Cleared $P = 0.047$; ^mixed-effects model fixed factors interaction, Maintain vs. Suppress $P = 0.001$, Replace vs. Suppress $P = 0.002$. Source data are provided as a Source Data file.

**Behavioral consequences of removal.** In the neuroimaging study we specifically did not want to query an individual's memory of an item after it was manipulated. We made this design decision because inserting such a query as that would have precluded us from examining how the removal operations influenced encoding on the subsequent trial (i.e., that query would have been an intervening and confounding event). Hence, a separate behavioral study was designed to examine how each of the removal operations influences the ability to make a subsequent WM decision about an item that was manipulated. We evaluated the behavioral consequences of the three removal operations by measuring the RT to recognition probes after each operation (Fig. 6; see the "Methods" section). Briefly, on each trial, two target pictures were encoded for 2.76 s. Next a cue appeared for 1.76 s indicating how one of the items (which randomly varied between the left and right) was to be manipulated, followed by a delay period of 1 s. Next a recognition probe lasting 1 s was shown and participants responded as to whether the probe had been presented at the beginning of the trial. Half of the probes consisted of a target item (either the manipulated or the non-manipulated item) and required a Yes response, while the other

half consisted of a foil from a recent trial or a novel stimulus, both of which required a No response. Task instructions made explicit that participants should respond Yes to items that belonged to each trial regardless of whether or not the item was manipulated. We excluded participants ($N = 50$) with accuracy lower than 75% on any trial type to ensure that we did not analyze data from participants who may have been confused about the task. The remaining participants ($N = 208$) performed exceedingly well on the task (~95% accuracy), and no statistically significant differences in accuracy were noted across conditions. Only data from positive probes (i.e., those requiring a Yes response) are analyzed in the current study. As expected, in the maintain condition, RTs for correct Yes responses were faster on recognition probes of items that were maintained ($T(70) = -2.47$, $P = 0.016$, $d = 0.29$, 95% CI $[-30.92, -3.32]$), compared to non-manipulated items from the same set of trials. This pattern was also observed for items that were designated to be replaced ($T(207) = -2.47$, $P = 0.014$, $d = 0.17$, 95% CI $[-16.44, -1.85]$), compared to baseline items that were not manipulated in that condition. This effect, however, was reversed for suppressed items, which yielded slower RTs relative

to baseline ($T(69) = 2.30$, $P = 0.025$, $d = 0.27$, 95% CI [2.28, 32.32])), suggesting that indeed suppressing representations of these items made them more difficult to access. There was no distinction between the two items on clear trials (as both items were to be cleared), so this within-condition comparison was not possible for cleared items. In addition, there were no differences for non-manipulated items across the other three conditions $F(2, 184.18) = 0.126$, $P = 0.881$, $\eta^2 = -7.4\mathrm{e}{-07}$), indicating that the baselines against which these effects were calculated did not differ between conditions. Nonetheless, there was a significant difference between manipulated items across conditions ($F(3, 347.98) = 5.211$, $P = 0.002$, $\eta^2 = 0.01$), with planned pairwise comparisons demonstrating a significant difference for maintain vs. suppress ($T(133.94) = -2.30$, $P = 0.023$, $d = 0.39$, 95% CI [$-58.56$, $-4.40$]) and suppress vs. clear ($T(122.53) = 2.01$, $P = 0.047$, $d = 0.31$, 95% CI [0.33, 49.43]), but no difference for maintain vs. replace ($F(1, 101.5) = 2.92$, $P = 0.09$, $\eta^2 = 0.001$, 95% CI [$-2.14$, 30.99]). Additionally, there were significant interactions between operation and probe type (manipulated, non-manipulated) on RTs for suppress vs. maintain ($F(1,139) = 11.35$, $P = 0.001$, $\eta^2 = 0.01$) and suppress vs. replace ($F(1,345.06) = 9.30$, $P = 0.002$, $\eta^2 = 0.005$]). Hence, maintaining an item in WM or having it targeted for replacement with another item allows participants to more quickly endorse having encoded the item on the current trial. Importantly, only suppressing an item slows the recognition judgment for that item. Combined with the neural finding that only suppression reduces proactive interference, this finding suggests that suppression produces the most effective removal of information from WM. Together with the neural results showing a selective reduction in proactive interference from suppression (Fig. 5), the selective slowing of recognition for suppressed items demonstrates the precise impact that suppression has on the contents of WM. This contrasts with more broad consequences from the suppression of episodic memory retrieval, which can produce forgetting of unrelated experiences in close temporal proximity to the time of suppression[37]. Understanding the relationship between these unique forms of memory suppression is an exciting area of ongoing research.

## Discussion

Altogether, this study demonstrates that applying multivariate analyses to brain activity patterns allows researchers (a) to verify that people have indeed removed information from mind, (b) to identify the consequences to the representation of the information being removed, and (c) to assess its impact on future processing. Whereas switching to a new thought or clearing the mind of all thoughts will reduce the attentional focus on the unwanted item, only by deliberately suppressing that item will its representational shadow be removed from the WM system thus freeing up those neural resources for encoding new information into WM.

These results have important implications for understanding basic cognition. First, they demonstrate that there at least three different methods of removing information from WM—replace, suppress, and clear—that have distinct effects on the representation of the information being manipulated. Second, as discussed above, the subsequent encoding fidelity results suggest that information in WM may be either temporarily removed from the focus of attention, which still incurs a cost via proactive interference on subsequent encoding, or more permanently removed from WM, thereby eliminating proactive interference and facilitating new learning. Third, they also suggest the possibility that different levels of representations for an item in WM—e.g., specific item information and general category information—may be

differentially affected by cognitive control operations (Fig. 4b), a dissociation which has been previously demonstrated in the updating of episodic memories[38,39].

These results are also likely to have important implications for understanding the etiology and treatments of psychiatric disorders. While intrusive thoughts are generally associated with PTSD—and are captured by the intrusions cluster of the diagnostic criteria[40]—they occur in many psychiatric disorders. Depression is characterized by intrusive negative thoughts and memory[41], anxiety by repetitive intrusive concerns about future negative events[42], obsessive-compulsive disorder (OCD) by thoughts of contamination and/or harm to self or others[43], and schizophrenia by intrusions of semantic and sensory information[44]. Hence one question for future research is the degree to which the cognitive operations examined in the current study are impaired or disrupted in such populations.

In addition, our results have implications for considering the degree to which suppression of thoughts is disadvantageous. For example, in PTSD suppression of the long-term memory of the traumatic event is thought to preclude it from being manipulated or modified to lessen the linkage of it and its attributes (e.g., loud noises) to fear. A common intervention, exposure therapy, is designed to bring the thought to mind so that the linkage of emotional information (e.g., fear) to information in episodic or semantic memory can be altered or re-structured[45].

Our results suggest that considering the role of WM processes in such disorders may also be fruitful. Once a memory has reached consciousness and entered WM, suppressing a thought may be advantageous, as this process allows for the complete removal of the information from mind. In comparison clearing or replacing that information may only temporarily shift attention away from it, with a trace of the information still existing in memory[4]. These findings might suggest a tiered approach to interventions regarding control over thought. It may be that training an individual to shift their attention by redirecting (i.e., replacing) thought or through mindfulness techniques (i.e., clearing thoughts) could be a fruitful first step followed by training to exert cognitive control to suppress the thought, and thereby reduce its potency. Further work will need to explore these ideas, but our results, nonetheless, point the way to potentially fruitful translation to clinical practice.

## Methods

**Participants**. A total of 60 participants (20 male; age, $M = 22.97$, SD = 4.77, handedness: right = 60) were recruited from the Boulder, CO, area for the fMRI study. Five participants failed to remain awake throughout all phases of the study and were excluded. An additional five participants were excluded due to poor fMRI classifier performance for the four cognitive operations in the central study (chance-level performance on AUC = 0.5), which indicates lack of engagement or inability to perform the different mental operations. The remaining 50 participants (17 male; age, $M = 23.52$, SD = 4.93) were included in all analyses. All participants had normal or corrected-to-normal vision, provided informed consent, and were compensated $75. The study was approved by the University of Colorado Boulder Institutional Review Board (IRB protocol # 16-0249).

A total of 259 participants (166 female; age, $M = 19.25$, SD = 2.15) took part in the behavioral study. One participant was excluded due to equipment failure, and 50 participants were excluded due to poor task performance (<75% accuracy for positive and/or negative), resulting in a final sample of 208 participants (70 Suppress, 67 Clear, 71 Maintain, 208 Replace; 135 female; age, $M = 19.17$, SD = 1.34). All participants had normal or corrected-to-normal vision, provided informed consent, and reported no history of brain injury, neurological or psychiatric disorder, nor severe cognitive or psychological problems. The study was approved by the University of Colorado Boulder Institutional Review Board (IRB protocol #18-0571). Participants were collected via convenience sampling as psychology students participating for course credit at the University of Colorado Boulder. Power analyses indicated that small effect sizes of 0.2, 0.3, and 0.4 could be detected with 0.8 power and an alpha of 0.05 with sample sizes of 277, 126, and 73 participants, respectively. As we expected to lose subjects due to stringent quality assurance procedures, we enrolled as many participants as possible. We determined that our final sample size ($N = 208$) was adequate to detect small effects, and the sample size was consistent with, or larger than, similar studies[33,37].

**Stimuli**. Stimuli for the fMRI study consisted of colored images (920 × 920 pixels) from three categories with three subcategories each: faces (actor/ musician, / politician), fruit (apple, grape, pear), and scenes (beach, bridge, mountain). Faces were recognizable celebrities and scenes were recognizable locales (e.g., a tropical beach) or famous landmarks. Images were obtained from various resources including the Bank of Standardized Stimuli[46] and Google Images. Six images from each subcategory were used, for a total of 18 images per category and 54 images in total. All images were used for both the localizer and study phases of the experiment.

Stimuli for the behavioral study consisted of colored images of familiar faces (e.g. Ellen DeGeneres) and scenes (e.g. Golden Gate Bridge) that were obtained from various resources including Google Images. There were 252 unique items per category (504 images total). Importantly none of the images was ever repeated across trials. To prevent stimulus-specific effect (e.g., familiarity), the images were fully randomized across trials, conditions, and participants. For replace trials, a subset of the images was used as replacement items (i.e., 24 items per category) and this set was consistent across participants.

**fMRI procedure**. The experiment consisted of two phases completed in order: a functional localizer and a central study. Prior to completing both tasks in the MRI scanner, participants received training on the tasks outside of the scanner, including nine trials of the functional localizer task (three of each category of stimuli) and four self-paced trials of the central study (one trial per condition). Both tasks involved presenting participants with the same set of color images, though the tasks differed in what the participants were asked to do when presented with these images. All stimuli were presented on a black background with task-related words and fixation crosses shown in a white font. All stimuli were presented via E-Prime (version 2.0.10.356)[47].

The functional localizer task allowed for the characterization of multivariate patterns of brain activity associated with attending to the different categories and subcategories of visual stimuli. Participants were presented with images, one at a time, and asked to rate the desirability of each image on a four-point scale. If the image was a face, participants were asked to rate "How much you would like to meet this person?", if the images was a fruit, participants were asked to rate "How much would you like to eat this fruit?", and if the image was a scene, participants were asked to rate "How much would you like to visit this place?". The first 7 participants to complete the study made category judgments on these stimuli instead. Participants were asked to make these ratings to promote attention towards the stimuli. Note that the same 54 images were used in each operation, so there was no bias of preference rating or stimulus-specific effects in any of the operations. This design choice precluded obtaining behavioral data on specific items. Additionally, to promote encoding of these stimuli, we informed participants that they would be asked to recall the images during the subsequent study phase. Across the entirety of the localizer task, participants completed five runs (6.17 min each, 30.85 min total) for a total of 270 trials, 90 trials of each category, 30 trials of each subcategory, with five trials of each image exemplar. A trial consisted of 3 TRs (1.38 s) of a given image, followed by a jittered inter-trial interval ranging between 5 and 10 TRs (2.3–4.6 s), consisting of a white fixation cross on a black background. Trials were grouped into subcategory-specific triplets (e.g., three actor/face image trials in a row), with each image exemplar shown once per run, resulting in six subcategory-specific triplets for each category within a single run. Each triplet was followed by a 13 TR (5.98 s) fixation block and each run began with 13 TR (5.98 s) instruction reminder screen. The order of subcategory-specific triples was optimized for BOLD deconvolution using optseq2[48].

The central study was designed to allow us to track the representational status of a WM item while it was being manipulated using five distinct cognitive operations: maintaining an image in WM (maintain), replacing an image in WM with the memory of an image from a different subcategory of the same superordinate category (e.g., replacing an actor with a politician; replace subcategory), replacing an image in WM with the memory of an image from a different category (e.g., replacing an actor with an apple; replace category), suppressing an image in WM (suppress), and clearing the mind of all thought (clear). Note that results for replace subcategory and replace category trials were nearly identical, and thus only replace subcategory data are presented in the main paper. We focused on category-level neural decoding (subcategory-level decoding was insufficiently powered) and replace subcategory trials were not suited for this analysis. For this reason, and to avoid biasing the operation classifiers, we excluded replace subcategory data rather than combining it with replace category data. On each trial (see Fig. 1a), participants were presented with an image for 6 TRs (2760 ms) followed by another 6 TRs of an operation screen instructing participants how to manipulate the item in WM, and then a jittered inter-trial fixation lasting between 5 and 9 TRs (2300–4140 ms), consisting of a white fixation cross centered over a black background. The operation screen consisted of two words in the top and bottom halves of the screen, presented over a black background. For the maintain, suppress, and clear operations, the two words were the same: maintain, suppress, or clear, respectively. In the two replace conditions, the word in the top half was switch, whereas the word in the bottom half indicated the subcategory of image that the participant should switch to (e.g., apple). During practice and before the beginning of the task, participants were instructed to only switch to thinking about an image that had previously been shown during the functional localizer

task. For example, if a participant was instructed to switch to thinking about an apple, that apple should be one of the apples that was presented to them during the localizer. Participants completed 6 runs of this task (9.01 min each, 54.05 min total), resulting in a total of 360 trials: 72 trials per operation, of which 24 trials were image category-specific trials per each operation condition. Each run had 40 TR long (18.4 s) fixation blocks at the beginning and end of the run. Within a single run, 12 trials were presented for each of the five operations, resulting in a total of 60 trials per run. Each image exemplar appeared at least once per operation condition across the entirety of the task. Trials were ordered pseudo-randomly within runs, with the order of trials optimized for BOLD deconvolution using optseq2[48].

**Behavioral procedure**. The behavioral study design consisted of a mixed within- and between-subjects design to measure the effects of the different WM operation conditions with unique items across trials. Participants were assigned to one of three groups that differed in the WM operation required: maintain, suppress, or clear. All participants additionally completed trials with the replace operation. Prior to the main task, participant completed a familiarization task for the replacement items to ensure that the items could be retrieved based on their names. In this familiarization task, on each trial, a single image appeared at the center of the screen along with two item–name options below the image for 4 s. Participants responded to the name that matched the image using their left or right index finger.

In the main task, participants in each group performed two blocks of trials, one requiring their group-specific operation (e.g., maintain) and the other requiring the replace operation. Order of operations was counterbalanced across participants (i.e., half of participants completed the replace condition first). Participants received practice trials prior to each block. Each operation consisted of 72 trials for a total of 144 trials, and the task lasted ~30 m. On each trial (see Fig. 6a), two items (i.e., one face and one scene) were presented for 2.76 s, one to each side of a central fixation cross with position counterbalanced, followed by an instruction screen for 1.76 s with an operation cue on one side of the central fixation. The location of the cue indicated which item had to be manipulated with the given operation. This item is the manipulated item while the un-cued item is the non-manipulated item. Cues for the clear operation appeared in the middle of the screen as both items were to be cleared from WM. The cues appeared in different colors and shapes depending on operations as following: green O for maintain, blue <item name of the replacement item> for replace, red X for suppress, and orange cloud for clear. Position of the cue for the maintain, replace, and suppress conditions varied randomly between the right and left. A 1-s retention interval followed with a central fixation cross. Afterwards, a probe picture was presented at the center of the screen for 1 s. Participants had to indicate, within 2.5 s, whether the probe had appeared at the beginning of the trial (regardless of whether the item was manipulated or not) with their right or left index fingers, with response mappings counterbalanced across participants. The next trial began after a 1.5 s blank inter-trial interval. Valid probes appeared on 50% of the trials, which consisted of either a manipulated or non-manipulated item. Negative probes consisted of 66.6% items seen during the previous trial and 33.3% novel items. All stimuli were presented via Psychopy2 1.93.1 using Python3.

**Data acquisition**. MRI data were acquired on a Siemens PRISMA 3.0 Tesla scanner at the Intermountain Neuroimaging Consortium on the campus of the University of Colorado Boulder. Structural scans were acquired with T1-weighted sequence, with the following parameters: repetition time (TR) = 2400 ms, echo time (TE) = 2.07 ms, field of view (FOV) = 256 mm, with a 0.8 × 0.8 × 0.8 mm³ voxel size, acquired across 224 coronal slices. Functional MRI (fMRI) scans for both the functional localizer and central study were acquired using a sequence with the following parameters: TR = 460 ms, TE = 27.2 ms, FOV = 248 mm, multiband acceleration factor = 8, with a 3 × 3 × 3 mm³ voxel size, acquired across 56 axial slices and aligned along the anterior commissure-posterior commissure line. For the functional localizer task, five runs were acquired in total, with each run consisting of 805 echo planar images (EPI), for a total of 4025 images across the five runs. For the central study, six runs were acquired in total, with each run consisting of a 1175 EPIs, for a total of 7050 images across the six runs.

**Regions of interest**. To characterize the WM representations of the images used in this study, we focused analyses on the ventral visual stream (VVS) in the occipi-totemporal lobes[49,50]. This mask consisted of the following anatomically defined regions, derived from the Harvard-Oxford Cortical atlas and thresholded at a 20% probability: intracalcarine cortex, lingual gyrus, lateral occipital cortex (inferior), occipital fusiform gyrus, occipital pole, parahippocampal gyrus (anterior and posterior divisions), temporal fusiform cortex (anterior and posterior divisions), temporal occipital fusiform cortex, inferior temporal gyrus (posterior and temporooccipital), middle temporal gyrus (posterior, anterior, temporooccipital), superior temporal gyrus (posterior and anterior), and temporal pole. To construct this mask for each participant, thresholded masks of these regions (bilaterally) were summed together and converted to individual's native brain space. The whole-brain mask consists of only gray matter that was segmented based on a high-resolution structural brain image using FMRIB's automated segmentation tool (FAST) provided in FSL. The ROI masks were then binarized so that voxels within

the mask had a value of 1 and voxels outside of the mask had a value of 0 (VVS: $M = 13,825$, SD = 1514 voxels; whole brain: $M = 37,343$, SD = 2962, Fig. 1b).

**Univariate fMRI analyses.** fMRI preprocessing and analyses were carried out using the FSL suite (version 5.0.10) (http://fsl.fmrib.ox.ac.uk). The first 10 EPI volumes of each run were discarded to allow the MRI scanner to reach steady-state stability. Preprocessing included motion correction via ICA-AROMA (version 0.3 beta)[51], an independent component analysis method for removing motion, high-pass filtering (100 s), and BET brain extraction[52]. Registration of EPI images into subject- and standard-spaces was executed using FLIRT (version 6.0)[53]. Individual subject EPI images were registered to that subject's MPRAGE structural image via linear Boundary-Based Registration[54] and then registered to the MNI-152 template via 12 degrees of freedom linear transformation. The resulting EPI images were smoothed using an 8 mm full-width half-maximum Gaussian smoothing kernel. FEAT (version 6.00) was used to model effects during the 6 TRs (2760 ms) during which the manipulation of the item was occurring, with fixation blocks at the beginning and end of each run serving as a baseline. The TRs during the presentation time of the stimulus and the TRs during the inter-trial interval fixations both served as EVs of no interest. Following Banich et al. (2015)[15], we examined three main contrasts. We used a voxel-wise threshold of $P < 0.0025$ after permutation testing of 10,000 iterations (calculated via Randomise (version 2.9) in FSL) as well as a cluster-thresholding which was corrected for multiple comparisons by using the null distribution of the maximum (across the image) cluster mass. Results are presented in Supplementary Fig. 1 and Supplementary Tables 5–9.

**Multivariate pattern classification.** FSL (version 5.0.8) (http://fsl.fmrib.ox.ac.uk) was used to preprocess the fMRI data. Functional volumes were corrected for motion, aligned to the mean volume of the middle run, temporal high pass filtered (128 s), and detrended. Timepoints with excessive motion were removed (framewise displacement, threshold = 0.9 mm[55]; $M = 11.4$ TRs removed, SD = 17.8). Before further analysis, the first 10 TRs of each run were trimmed to remove unstable signals. The Princeton MVPA toolbox (www.pni.princeton.edu/mvpa) for MATLAB (2019b) was used for all within-subject fMRI pattern classification analyses[18,20,56] with L2-regularized, non-multinomial (one-vs.-others, for each category) logistic regression. All classifiers were trained and tested within each participant in their native brain space. Two sets of fMRI classifiers were built: (1) WM representation classifiers trained on functional localizer task data, and (2) WM operation classifiers trained on central study data. To validate classifier performance, $k$-fold leave-one-out cross-validation was performed across all runs: five runs of localizer data for WM representation classifiers ($M = 4.86$ runs, with a few runs missing across participants), and six runs of study data for WM operation classifiers ($M = 5.72$). Feature selection was performed for each training set using a voxel-wise ANOVA across classes (threshold: $P = 0.05$) with the regressors shifted forward 4.6 s (10 TRs) to account for hemodynamic lag. To find the optimal L2 penalty value for each classifier's best fitting model, the cross-validation was done with different penalties in two steps: (1) eight iterations with a broad range of penalties (from 0 to 10,000 with exponential increase) and then (2) 10 iterations in a narrow range around the best penalty value from the first step. A single penalty was chosen for each subject based on the maximum generalization performance from this iterative penalty search.

The WM representation classifiers were trained on preprocessed data from the functional localizer task. Data from the VVS ROI were used to build category-level classifiers (3 classes: face/fruit/scene) and whole brain data were used to build subcategory-level classifiers (9 classes: actor/musician/politician/apple/grape/pear/beach/bridge/mountain). Trial regressors were modeled with a mini-block boxcar (3 trials from a single subcategory per mini-block, 14.26 s (31 TRs)) and shifted forward 4.6 s (10 TRs) to account for hemodynamic lag. Across the cross-validation folds, the feature-selected voxels contained 55.83% of original voxels under each individual's VVS ROI for category ($M = 7747$, SD = 1642) and 69.93% of whole-brain mask for subcategory ($M = 26,145$, SD = 5441). Classifier accuracy was obtained from 5-fold cross-validation across localizer runs using the optimal penalty for each participant (category: $M = 1196.81$, SD = 4427.01; subcategory: $M = 194.14$, SD = 806.59). To verify the accuracy of the classifier, the one-sample $T$-test was conducted for each class (i.e., category or operation). We used an alpha level of 0.05 with two-tailed for all statistical tests. The accuracy of the WM representation classifiers were reliably above chance at the category level (averaged across categories: $M = 0.81$, SEM = 0.013, chance = 0.33, more reliable than $T(49) = 27.02$, $P < 0.001$, $d = 3.822$, 95% CI [0.39, 0.45]) and at the subcategory level ($M = 0.33$, SEM = 0.012, chance = 0.11, $T(49) = 8.77$, $P = 1.31e{-}11$, $d = 1.240$, 95% CI [0.12, 0.19], Fig. 3a, Supplementary Table 2). Classifier area under the receiver operating characteristic (ROC) curve (AUC) scores were significantly above baseline (0.5) at the category level ($M = 0.91$, SEM = 0.009, $T(49) = 33.14$, $P < 0.001$, $d = 4.687$, 95% CI [0.35, 0.4]) and at the subcategory level ($M = 0.75$, SEM = 0.010, $T(49) = 12.21$, $P = 2.22e{-}16$, $d = 1.727$, 95% CI [0.17, 0.24]). Additionally, the WM representation classifier was replicated with a single penalty value of 50 across participants, and the classifier accuracy and sensitivity remained reliably above chance at the category level (accuracy: $M = 0.80$, SEM = 0.014, $T(49) = 26.74$, $P < 0.001$, $d = 3.781$, 95% CI [0.38, 0.45]; AUC: $M = 0.91$, SEM = 0.009, $T(49) = 32.76$, $P < 0.001$, $d = 4.632$, 95% CI [0.35, 0.4]) and at the subcategory level (accuracy: $M = 0.32$, SEM = 0.012, $T(49) = 8.43$, $P = 4.27e{-}11$,

$d = 1.192$, 95% CI [0.12, 0.19]; AUC: $M = 0.75$, SEM = 0.010, $T(49) = 12.18$, $P = 2.22e{-}16$, $d = 1.723$, 95% CI [0.17, 0.24]).

Data from all localizer runs were then used to re-train the WM representation classifiers and decode the central study data. Training was done with an individualized optimal penalty derived from the cross-validation analysis, and a new feature selection was performed (category: 57.81% of the original voxels; $M = 8021$ voxels, SD = 1653; subcategory: 71.79% of the original voxels; $M = 26,615$ voxels, SD = 5268). These classifiers were reliably accurate at the category level ($M = 0.80$, SEM = 0.012, more reliable than $T(49) = 28.16$, $P < 0.001$, $d = 3.983$, 95% CI [0.35, 0.4]) and the subcategory level ($M = 0.28$, SEM = 0.008, $T(49) = 9.7$, $P\mathrm{s} < 5.48e{-}13$, $d = 1.372$, 95% CI [0.08, 0.12], Supplementary Table 2). These classifiers were used to decode every timepoint to construct trial-averaged decoding time series. A 13.8 s time window (30 TRs, unshifted, from the onset of each trial) was used to evaluate the trajectory of the average classifier evidence for the WM item's category in each condition (Fig. 4a). The data were baseline corrected by removing the mean target classifier evidence, separately for each condition, from the first 2.76 s (6 TRs) from all subsequent time points. This procedure had no effect on any statistical comparisons between conditions but centered the data at trial onset to 0 rather than 0.45 (classifier evidence, ranging from 0 to 1). To highlight the removal of information from WM, we then recoded these data using the classifier evidence from maintain as a baseline by subtracting these values from the classifier evidence values for the three removal conditions (Fig. 4b). For statistical tests, we focused on a 6.9 s (15 TR) time window beginning at the onset of the operation (TR 7) through the end of the longest fixation period (TR 21). This analysis window was then segmented into five contiguous blocks (1.38 s [3 TRs] per block) and a repeated-measure one-way ANOVA and pair-wise $T$-tests with false discovery rate (FDR) for multiple comparison correction was applied to the averaged target category classifier evidence scores in each block. To identify the removal start point for each condition, one-sample $T$-tests were used in each block to compare the classifier evidence scores against zero.

The WM operation classifiers were trained on preprocessed data from the central study. Trial regressors for the five operations (maintain/replace subcategory/replace category/suppress/clear) selected data on each trial during the operation period (2.75 s, 6 TRs) and the subsequent fixation period (jittered from 2.3 to 4.41 s, 5–9 TRs) and shifted forward 4.6 s (10 TRs) to account for hemodynamic lag. We included the fixation period in the regressor because we found informative signals from a separate analysis in which the classification was trained with small training time window (2.3 s, 5 TRs) to decode the middle time point in that window (sliding-time-window classification). For example, the signal on TR 3 was decoded with the classifier trained with {1–5} TRs window. The training window was slid to decode the time points from the onset of each trial to the end of the longest trial period (14.2 s, 31 TRs), when the regressor was shifted forward by 4.6 s to account for hemodynamic lag. The WM operation classifier with sliding-time-window was significantly high for all five operations in the fixation period after adjusting for hemodynamic lag (10.58–14.2 s, 23–31 TRs; averaged across conditions: $M = 0.29$ with chance level of 0.2, SEM = 0.011; one-sample $T$-test, more reliable than $T(49) = 4.76$, $P = 1.78e{-}05$, $d = 0.673$, 95% CI [0.03, 0.08]). To verify that the classifier accuracy in the fixation window, which was partially overlapped with the next trial, reflects only the operation signals from the current trial, we removed back-to-back trials that involved the same operation, and the results were consistent ($M = 0.29$, SEM = 0.011, more reliable than $T(49) = 4.31$, $P = 7.87e{-}05$, $d = 0.609$, 95% CI [0.03, 0.08]).

In the WM operation classifiers, the feature-selected voxels were 59.60% of each individual's whole-brain mask ($M = 22,325$, SD = 5052). Classifier accuracy was obtained from the 6-fold leave-one-run-out cross-validation with individualized optimal penalty ($M = 1459.22$, SD = 338.67) derived iteratively as described above. The classifiers were reliably accurate ($M = 0.42$, SEM = 0.016; one-sample $T$-test vs. chance (0.2), more reliable than $T(49) = 10.27$, $P = 8.28e{-}14$, $d = 1.452$, 95% CI [0.18, 0.27]), and sensitive (AUC: $M = 0.72$, SEM = 0.013; one-sample $T$-test vs. chance (0.5), $T(49) = 12.83$, $P < 0.001$, $d = 1.815$, 95% CI [0.17, 0.24], Supplementary Table 1) across all operations. We found that the replace subcategory ($M = 0.34$, SE = 0.010) and replace category ($M = 0.33$, SE = 0.009) was not distinguishable from each other at the operation level (paired $T$-test for target vs. nontarget, less reliable than $T(99) = 1.42$, $P = 0.16$, $d = 0.142$, 95% CI [−0.004, 0.03] for accuracy and evidence), thus we removed the replace-subcategory condition from the main results. With the regressors for the four operations (maintain/replace/suppress/clear), the voxels identified from feature selection were 54.55% ($M = 20,431$, SD = 4882) of the whole-brain mask. The cross-validation classification with optimal penalty ($M = 653.27$, SD = 2546.52) also showed reliable accuracy ($M = 0.51$, SEM = 0.021, more reliable than $T(49) = 10.26$, $P = 8.55e{-}14$, $d = 1.451$, 95% CI [0.2, 0.3], Fig. 2a) and sensitivity (AUC: $M = 0.74$, SEM = 0.016; $T(49) = 12.25$, $P = 1.11e{-}16$, $d = 1.733$, 95% CI [0.17, 0.24], Supplementary Table 1) across all operations. Paired $T$-tests were applied to test target vs. non-target categories of classifier predictions for the comparisons between suppress vs. clear and replace vs. the two removal operations. Finally, we replicate the cross-validation with a single penalty value (penalty = 50), and the classifier were still reliably accurate and sensitive (accuracy: $M = 0.51$, SEM = 0.021, more reliable than $T(49) = 10.07$, $P = 1.62e{-}13$, $d = 1.424$, 95% CI [0.18, 0.26]; AUC: $M = 0.74$, SEM = 0.016, $T(49) = 12.16$, $P = 2.22e{-}16$, $d = 1.72$, 95% CI [0.17, 0.24], Supplementary Table 1). The classifier confusion matrix for all classifiers (Figs. 2a and 3a) were generated with Python seaborn v0.8.0[57].

To identify brain areas that contributed to the identification of each operation, we generated classifier importance maps[21]. For each individual, positive and negative importance maps were generated in subject native space. Positive maps for an operation consisted of voxels whose: (a) mean $z$-scored fMRI activity for that operation was positive, and (b) classifier weight was positive. These voxels contributed to the identification of the operation when their activity was higher than average. Negative maps consisted of voxels whose: (a) mean $z$-scored fMRI activity for that operation was negative, and (b) classifier weight was negative. These voxels contributed to the identification of the operation when their activity was lower than average. Per the method described in McDuff et al.[21], voxels whose mean fMRI activity and weight had opposing polarities were assigned an importance value of zero and were ignored. Individual importance maps were then $z$-scored, normalized to the Montreal Neurological Institute (MNI, $3 \times 3 \times 3$ mm$^3$) template. When $z$-scoring, both positive and negative values combined as absolute values to normalize the values within a single distribution and were then separated to their original maps. These maps were then combined across subjects and the top 5% of importance values were selected for the group-level positive maps and the negative maps. These maps were then corrected with cluster-extent thresholding (10 voxels) and smoothed with a 12 mm FWHM Gaussian kernel using FSL and visualized using FreeSurfer (version 5.3)[58] (Fig. 2b).

**Classification between subjects**. The BrainIAK toolbox[59] with Python was used for between-subject pattern classification analyses with L2-regularized (penalty = 50), non-multinomial (one-vs.-others, for each category) logistic regression. All data from all participants ($N = 50$) were normalized to MNI standard brain space and concatenated, so that all voxels are anatomically aligned across participants. The first half of all runs was used for feature-selection, and the other half of the data was used for training and testing the classifier. One participant was excluded from this analysis due to missing half of their central study data, thus $N = 49$ participants contributed to this analysis. The top 10,000 voxels were first feature-selected from the whole-brain gray mask segmented from standardized MNI brain (17.13% of 58,229 voxels, ANOVA threshold: $P < 0.001$), and the feature dimensions were then reduced to 70 components using principal component analysis (PCA) provided in BrainIAK. This number of components was selected as the optimal value to maximize classification accuracy using a repeated cross-validation testing scheme. The same feature-selected voxels of the testing data were transformed to component space and used for $N$-fold leave-one-participant-out cross-validation across all participants, with the operation regressor shifted forward 4.6 s to account for hemodynamic lag.

We also conducted another version of between-subject classification with functionally aligned data using the hyperalignment procedure[60] in BrainIAK. The first half of the data was used for feature selection (top 5000 voxels per hemisphere in each individual's native brain space, 26.71% of $M = 37,436$ voxels, ANOVA threshold: $P < 0.005$) and hyperalignment, and the other half was used for classification. To create the template (i.e., common space) for each hemisphere to which all participants' voxels were aligned, we performed two steps. First, each individual's voxels were aligned to a reference, which was the first participant with a full dataset, using Procrustean transformation. Then, for each step in this iterative procedure, the reference was updated by averaging the current reference with the newly aligned participant. In the second step, each individual's voxels were aligned again to the reference that was obtained from the completion of the first step. Then, all individual's features in the common space were averaged, and this group-averaged feature set served as the final reference template. Finally, we obtained transformation parameters for each participant, using this final template, and then transformed the feature-selected voxels of testing data into the hyperaligned common space. The hyperaligned features were then reduced using PCA (70 components, which was also the optimal value) and then used for cross-validation classification. The classifier accuracy was reliably above chance for both of the anatomically aligned ($M = 0.403$, SEM = 0.012, one-sample $T$-test: more reliable than $T(48) = 6.4$, $P = 6.2e-08$, $d = 0.914$, 95% CI [0.06, 0.12], Fig. 2a, Supplementary Table 1) and hyperaligned between-subject classifiers ($M = 0.378$, SEM = 0.014, $T(48) = 4.9$, $P = 1.13e-05$, $d = 0.7$, 95% CI [0.05, 0.12]) across all operations.

**Representational similarity analysis**. To decode the neural representation of individual stimuli in the central study, we applied RSA[19] with custom code in MATLAB (2017a). Each stimulus in the central study was also viewed in the localizer task (5 exposures per item across 5 runs). We defined a template pattern of activity for each item (54 items total) from the localizer data and used this to identify item-specific representations in the study data. To choose voxels for this analysis, we performed a two-step procedure to identify category-selective voxels and then item-selective voxels for each stimulus. Within the VVS ROI, we modeled beta estimates using SPM12 for three categories (face, fruit, scene) in the localizer data with boxcar regressors on mini-blocks (3 trials per mini-block, 14.26 s: 31 TRs) and motion parameters in a general linear model (GLM), utilizing a canonical hemodynamic response function. To select voxels for each category within this mask, a target vs. non-targets contrast ($t$-contrast in SPM) was computed (e.g. face vs. {fruit, scene}). Voxels were selected that passed threshold (uncorrected $P < 0.05$) and cluster correction (voxel extent = 10). Across the three categories, the total

number of voxels selected were 15.18% of original mask ($M = 2125$, SD = 434; face: $M = 2164$, SD = 716; fruit: $M = 1757$, SD = 1007; scene: $M = 2453$, SD = 833).

The first round of feature selection for each item was to choose the appropriate category-specific voxels from this GLM. The second round involved weighting the voxels in an item's category-specific mask based on GLM fits for that item. This was done by specifying a unique regressor for each item in a single GLM (LS-A in ref. [61]). Item-specific beta estimates were computed by contrasting each item with the 53 other items (e.g., $t$-contrast: item 1 vs. {items 2…54}; $M = 0.69$, SEM = 0.013 $t$-contrast betas). Each item appeared once in each of the five localizer runs. The mean voxel activity pattern for an item was computed by averaging across the five repetitions and weighting it with the item-specific $t$-contrast values. These weighted item template patterns served as the reference to quantify the encoding fidelity (i.e., the pattern similarity) of these items in the central study. To verify the decodability of item-specific activity patterns in the localizer data, we correlated data across the five repetitions of all six exemplars for each stimulus subcategory (Fig. 3b). For RSA scores, correlation coefficients (Pearson $r$) were computed and then converted using Fisher's $z$ transformation for statistical analysis. Clustering of high-correlation scores along the diagonals of these matrices confirms that item-specific patterns were more similar to repeated presentations of the same item ($M = 0.3$, SEM = 0.015) than to other items within a single subcategory ($M = 0.09$, SEM = 0.007) across all subcategories (paired $T$-test, $T(49) = 24.78$, $P < 0.001$, $d = 3.504$, 95% CI [0.2, 0.23], Fig. 3b). Across all 54 items, the RSA scores for the same items were significantly higher than other items from the same category ($M = 0.08$, SEM = 0.006, $T(49) = 23.6$, $P < 0.001$, $d = 3.338$, CI [0.21, 0.25]) and other items from different categories ($M = 0.04$, SEM = 0.002, $T(49) = 20.54$, $P < 0.001$. $d = 2.905$, 95% CI [0.24, 0.29]) after Tukey–Kramer correction (one-way ANOVA, $F(2, 98) = 448.89$, $P = 4.57e-50$, $\eta^2 = 0.902$).

To decode the probability of the specific item signal during study data with RSA, the study pattern was also weighted with the item-specific $t$-contrast values (e.g., item 1 decoding = RSA between item 1 template from the localizer and the study pattern weighted with item 1's $t$-contrast beta). To calculate RSA scores between template patterns from the localizer data and central study weighted patterns, the correlation coefficient (Pearson $r$) was computed and then converted using Fisher's $z$ transformation for statistical analysis. The sensitivity of this analysis is verified in Fig. 5a which shows the mean correlation for items presented in the central study. These data correspond to the 2.76 s (6 TRs) of item presentation on each trial, shifted forward by 4.6 s (10 TRs) to account for hemodynamic lag. The RSA scores for the presented items (target, $M = 0.17$, SEM = 0.011) were statistically higher than other items from the same category as the target item (related, $M = 0.13$, SEM = 0.009, $T(49) = 12.49$, $P = 1.11e-16$, $d = 1.767$, 95% CI [0.03, 0.05]) and other items from different categories (others, $M = 0.02$, SEM = 0.004, $T(49) = 16.57$, $P < 0.001$, $d = 2.343$, 95% CI [0.14, 0.18]) after Tukey–Kramer correction (one-way ANOVA, $F(2, 98) = 263.88$, $P = 3.51e-40$, $\eta^2 = 0.843$). We refer to the target RSA score as the 'encoding fidelity' of a given item. This RSA procedure was then used to decode item-level activity patterns at each time point of the study data (Fig. 4b, bottom, Supplementary Table 3).

RSA was also used to evaluate the impact of the removal operations on subsequent encoding. We hypothesized that when a target item was removed from WM, this should reduce proactive interference for encoding a new item into WM. Moreover, this reduction in proactive interference should be greatest when the new item is related to the previous item, as these representations would share the most neural resources[30]. For each operation, we separated trials based on the correspondence between the category of the item on the current trial (item N) and the item on the next trial (item N + 1). We computed the encoding fidelity of each trial and compared same category trials (when items N and N + 1 are the same stimulus category) to different category trials. One-sample $T$-test and one-way ANOVA for repeated measures compared same-minus-different encoding fidelity scores for each operation separately and across operations with Tukey–Kramer correction for multiple comparisons. The encoding fidelity was only positive ($M = 0.041$, SEM = 0.009) and significantly higher for suppress than the other operations (all paired $T$-test were more reliable than $P = 1.11e-04$, $d = 0.669$, 95% CI = [0.04, 0.09], Fig. 5b, Supplementary Table 4). Across same and different categories, the number of trials varied (same: $M = 19.29$, SD = 4.32; different: $M = 48.09$, SD = 7.27 trials). In the replace condition, we removed the trials when the new-item, which acted as a replacement of the item N, and the item N + 1 are the same category (same–new-item; $M = 22.7$ trials out of $M = 45.6$ total, Fig. 5b) to isolate the effect from the new-item on the subsequent encoding. These trials were only subset of different category condition because new-item was always different category from the target item. Encoding fidelity for the items N + 1 were significantly lower for the same–new-item trials than the other trials in different categories ($T(49) = 2.29$, $P = 0.027$, $d = 0.323$, 95% CI [−0.002, 0.05]). Consistent with our hypothesis, this also suggests the proactive interference of the current information on the subsequent encoding.

In a separate analysis, we ran a bootstrap resampling analysis (1000 iterations) to replicate this RSA result (Fig. 5b) with the same number of trials across same vs. different conditions and operations (sample = 15 trials with replacement). The resampled data with replacement was normalized across conditions (Anderson–Darling normality test, $P$s > 0.052). Bootstrap results were consistent with our main results showing that the same-minus-different encoding fidelity score of item N + 1 for the suppress condition was significantly higher than other

operations ($Ps < 0.001$) with no differences for other operations to each other ($Ps > 0.317$). Additionally in the bootstrap analysis, we only used the removed trials in the replace condition as the same condition (i.e., new-item and next-item were the same category) and compared them with the trials in the different condition (e.g., trial $N$: Category 1 ⇒ switch to Category 2; trial $N + 1$: {Category 2 (for same category) or Category 3 (for different category)}). The results showed the same pattern as the original results, suggesting that the prior item in WM proactively interferes with the next item when they share categorical features (same < different, $P = 0.036$).

**Behavioral data analyses**. To measure the impact of each operation on the items being manipulated, reaction times (RTs) in the probe task were calculated for correct trials only. RTs below 200 ms or >2.5 standard deviations above the within-subject mean RT were excluded ($M = 1.49\%$ of trials, SEM = 0.09%). First, statistical tests were conducted on the individual mean RT values. Paired-sample $T$-tests were applied to compare recognition RTs for manipulated verses non-manipulated items for each operation (Fig. 6b). Mixed-effects models testing condition effects were run separately on RTs for manipulated and non-manipulated items. Because all participants completed the replace condition as well as one other condition, subject intercepts were included as a random factor. Follow-up pairwise comparisons were conducted for maintain verses suppress and for suppress verses clear (independent-samples $T$-tests) and for maintain verses replace (mixed effects model including subject as a random effect). No corrections were made for multiple comparisons. Mixed-effects models incuding subject as a random effect were also used to test fixed effect interactions between operation and manipulation (manipulated verses non-manipulated). All analyses were carried out in R (version 3.6.2). Mixed-effects models were conduced using lme4 (version 1.1-21) and $T$-tests were conduted using stats (version 3.6.2). Effect sizes for $T$-tests were calculated using the cohens_d function from the apa library (version 0.3.3). Effect sizes for mixed-effects model effects were determined using the DEE method described in Correll, Mellinger, and Pedersen (under revision)[62]. The results were visualized with the YaRrr[63] package in R.

**Statistics and reproducibility**. The fMRI experiment was performed once by each participant, and the behavioral experiment was performed once by a separate group of participants. No replication of either experiment was conducted.

**Reporting summary**. Further information on research design is available in the Nature Research Reporting Summary linked to this article.

## Data availability
Deidentified data are available from the authors upon request. Source data are provided with this paper.

## Code availability
All analysis code is available in GitHub. https://github.com/LewisPeacockLab/NCOMMS1930876B

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

## Acknowledgements

We thank Kathy Pearson for her help in computational and data management support. This work was supported by NIH R21MH108848 to M.T.B. and J.A.L.-P., and by NIH R01EY028746 to J.A.L.-P.

## Author contributions

M.T.B., J.A.L.-P., and H.R.S. conceived of and designed the experiment; H.R.S. and L.L.S. recruited participants and collected data; H.K. and L.L.S. analyzed data; All authors wrote the manuscript.

## Competing interests

The authors declare no competing interests.
