## [Peer Review File · Nature Communications]

Editorial Note: Parts of this Peer Review File have been redacted as indicated to remove third-party material.

Reviewers' Comments:

Reviewer #1:

Remarks to the Author:

The present study examined different potential mechanisms of removing information from mind. This is a topic that has long held prominence in psychology from people's failures to not think of a white bear, to the increased mental clutter caused by the lingering representations of irrelevant information (proactive interference). The ability to control one's thoughts has immense practical importance, but the most effective strategy for doing so, and its mechanistic underpinnings, have remained elusive. This manuscript provides a fascinating look into some of the neural mechanisms. I believe it will have widespread interest.

Using fMRI and sophisticated analysis techniques, the authors show that there are three distinct methods of ridding information from mind: replacing it with new information, actively attempting to suppress it from mind, and clear one's mind of thought. Each of these methods has distinct neural signatures. Moreover, each method has distinct consequences on information representation. Replacing irrelevant information with new information seems to be the most effective means of removing active signatures of the irrelevant information in the fMRI signal. Clearing is the next most effective, and suppressing is the least effective. However, all methods appear to reduce the representation of irrelevant information relative to active maintenance. Most strikingly, suppressing representations has a lingering effect on the encoding of new information. This may release the mind from proactive interference. If so, this is a truly novel and impactful result.

Major Concern

1) I have but one major concern, and it is major. While I take no fault in the methodology presented, it is striking that there are no behavioral data. From these data, it is unclear what effects these different removal mechanisms have on behavioral manifestations of memory itself. Absent behavioral data, one cannot infer whether these different neural mechanisms actually impact the cognitive system.

My guess is that the authors did have a surprise memory test at the end of the paradigm which may not have yielded conclusive results. Given the number of repetitions of each stimulus necessary for the techniques employed, such a result would be unsurprising. However, I would recommend that the authors perform a separate behavioral study (no imaging necessary) with trial unique stimuli nested within different categories. In that case, the authors can better examine whether the removal mechanisms, and suppression in particular, truly impact proactive interference.

Minor Concerns

1) On p. 8 the authors speculate "Somewhat paradoxically the trajectory for suppress is quite similar to that for maintain, suggesting that a representation must be within the focus of attention in order for it to be suppressed." The conclusion does not necessarily follow from the data. Moreover, if it did, I see no way in which it makes mechanistic sense. Note that the authors are interpreting BOLD activity which may or may not monotonically relate to neural activity. Active neural firing is not the only metabolically demanding phenomenon, and it is possible that the mechanisms underlying suppression may reduce neural firing, yet increase metabolic demand relative to some resting baseline. Therefore, I would encourage the authors to be a bit more agnostic in their interpretation giving the theoretical peculiarity of positing that a suppressed item is in the focus of attention.

2) On p. 9, the authors interpret differences in the first moment at which removal at the item-level is distinct from maintenance. Given that there are not significant differences among the removal conditions until far later, I do not believe interpreting differences among the conditions is warranted at the item-level. The differences in timing at the category-level appear much clearer and would appear to be directly related to the overall strength of the removal process.

3) In Fig 4A, the authors show the time course of the “new item” acting as a replacement in the replace condition. It would be useful to perform a surrogate analysis for the other removal conditions as a reference. That is, suppose that the authors randomly assigned a replacement category for the suppress and clear conditions. What would the time course of that surrogate look like? Given that evidence against one category is evidence for other categories, it would be appropriate to perform this analysis to demonstrate that the evidence for the new item in the replace condition really reflects the new item, and not simply the drop off of evidence for the old item.

4) In the abstract, the authors suggest that the only method to demonstrate that information is removed from mind is self-report. That is a bit overstated. For example, the work by Oberauer shows that lingering representations can be tracked using multiple methods such as the effect of those items on scanning (i.e. load effect) and slowed rejection of those items on probes (i.e. Monsell proactive interference effect). The authors might draw upon such methods for the behavioral study that I recommended. What is true is that the neural mechanisms remain poorly specified, as well as the mapping of neural mechanism to behavior, both of which are novel and impactful enough in themselves.

5) One matter that would be important to address with behavioral work is whether suppression is beneficial to future memory or detrimental. If it reduces proactive interference, that would suggest a beneficial effect. However, work by Anderson and colleagues (Hulbert et al., 2016) suggests a penumbra effect of suppression information from mind. It would good to reconcile these positions.

Reviewer #2:

Remarks to the Author:

The current study investigated working memory removal processes, comparing maintain, replace, clear and suppress instructions on neural representations. The results suggest that these cognitive operations are separable and have distinct effects on item representations and subsequent interference. I found the study impressive in many ways – the design is clean and straightforward, the study seems well powered and the analyses are thorough. I do have some concerns about methodological details, but my main reservation is related to the lack of behavioral measures to verify the instructional manipulation and the relationship between the neural representations and cognitive outcomes.

First, how do we know the manipulation is working as we would expect? There is no behavioral evidence and the differences between conditions in item decoding wouldn't be expected a priori. I.e., the magnitude of the item removal effect being greater in replace than clear than suppress could indicate a failure of the instructional manipulation. Was any behavioral data collected in this regard? The instructions should also be included.

While the WM operation decoding analysis suggests that people were indeed performing qualitatively different tasks in the 4 conditions rather than e.g., different degrees of something similar, there is still the possibility that this decoding is picking up on perceptual differences of the cue words. Some effort

could be made to remove the cue period from the decoding analysis and/or remove perceptually sensitive regions from the classifier. Another approach might be to test the classifier across participants, which should presumably be less sensitive to visual word differences between conditions. This could also help validate the instructional manipulation.

The subsequent proactive interference encoding fidelity analysis is a nice idea, but there are conceptual and interpretational concerns. Conceptually, while similarity could lead to interference, it is unclear whether category-level similarity is sufficient to cause interference here or whether there could even be the opposite -- a congruency benefit. Indeed, the authors argue that category-level congruency could be driving the post-suppression item representational enhancement effect.

Regarding the interpretation, the lack of a behavioral measure of success makes this quite challenging given that the neural results are not clearly consistent. The suppress condition shows the least evidence for removal in the decoding analyses, but then the least evidence of proactive interference on fidelity. These results are difficult to reconcile. One possibility is that these effects are driven by differences in difficulty, where "suppress" is the most difficult operation. That would explain the decoding effects and could potentially account for the interference analysis as well (see next comment). Is there any way to rule out such an explanation?

In figure 5b, same and different should be plotted separately. This is critical, because it is possible that differences in "fidelity" of the different categories is driving the effect of operation, which could undermine the argument that "suppress" is the most effective operation. For example, if "suppress" is the most difficult operation, it might interfere with the subsequent trial for different categories, which could enhance the same - different value. Of course, there should still be a control for overall similarity in the fidelity measure, but this is better captured by subtracting out similarity with other (related) non-target items.

The authors should examine whether reduced fidelity in the proactive interference analysis could be driven by category-level repetition suppression effects. That is, there may be a reduced overall neural response to an item from a category or subcategory that just appeared. Comparing the effects of reduced univariate activation to reduced item representational fidelity would inform the nature of the suppression.

The authors state that because "replace category" was indistinguishable from "replace subcategory," they removed the latter from the main results. I don't understand the logic here -- why not collapse these two conditions?

It's unclear how the optimal penalty was chosen for each participant. The description makes it sound as though the penalty with the best cross-validation performance was chosen but that would inflate cross validation accuracy (i.e., double dipping). Was there a separate held out set? In addition, it's unclear how this procedure would result in a single penalty per subject.

The authors should include non-baseline corrected timecourses in supplement -- why should there be any difference from 0 on average?

The regions that were used for each analysis should be included in the figures.

Lastly, because there are no behavioral outcome measures here, the level of discussion of potential therapeutic implications seems inappropriate.

Reviewer #1

Major Concern:

1. I have but one major concern, and it is major. While I take no fault in the methodology presented, it is striking that there are no behavioral data. From these data, it is unclear what effects these different removal mechanisms have on behavioral manifestations of memory itself. Absent behavioral data, one cannot infer whether these different neural mechanisms actually impact the cognitive system.

My guess is that the authors did have a surprise memory test at the end of the paradigm which may not have yielded conclusive results. Given the number of repetitions of each stimulus necessary for the techniques employed, such a result would be unsurprising. However, I would recommend that the authors perform a separate behavioral study (no imaging necessary) with trial unique stimuli nested within different categories. In that case, the authors can better examine whether the removal mechanisms, and suppression in particular, truly impact proactive interference.

Point taken. We agree with the reviewer that the absence of behavioral data is a limitation of our original neuroimaging study, which was designed to optimize our ability to track changes to working memory (WM) representations in the brain via multi-voxel techniques. For this purpose, we repeated each stimulus multiple times in the localizer (to obtain robust item-level representations), and across each of the experimental conditions (to minimize stimulus-specific variability across conditions). These design choices precluded us employing unique behavioral probes.

However, in response to the reviewer's concern, we have since conducted a separate behavioral study in which we focused on the immediate behavioral consequences of the cognitive operations used in this study. Specifically, we modified the "recent negatives" paradigm to investigate the behavioral impact (operationalized with response time, RT) of maintaining, replacing, clearing, and suppressing working memories of unique pictures similar to those used in the main study. The results demonstrate behavioral differentiations for these cognitive operations and thus they provide solid evidence that the cognitive operations used in our main fMRI study indeed have behavioral manifestations on memory. We have described the procedures for this experiment in the Supplementary materials, and we have included new behavioral results in the main manuscript along with a new figure (**Fig. 6**).

We have made the following additions to the manuscript to include these new findings:

(P. 25) [in discussing the design of our neuroimaging study] Note that the same 54 images were used for each operation, to avoid bias of preference rating or stimulus-specific

effects in any of the operations. This design choice precluded obtaining behavioral data on specific items.

(P. 16) In a separate experiment (**Fig. 6**), we evaluated the behavioral effects of these removal operations by measuring the response time (RT) to recognition probes after each operation (see **Supplementary Materials** for details). Briefly, on each trial, two target pictures were encoded for 2.76 s. Next a cue appeared for 1.76 s indicating how one of the items (which randomly varied between the left and right) was to be manipulated, followed by a delay period of 1 s. Next a recognition probe lasting 1 s was shown and participants responded as to whether the probe had been presented at the beginning of the trial. Half of the probes consisted of a target item (either the manipulated or the non-manipulated item) and required a “yes” response, while the other half consisted of a foil from a recent trial or a novel stimulus, both of which required a “no” response. Only data from positive probes (i.e. requiring a “yes” response) are analyzed in the current study. As expected, in the maintain condition, RTs for correct “yes” responses were faster on recognition probes of items that were maintained ($t(70) = -2.47$, $P = 0.016$), compared to non-manipulated items from the same set of trials. This pattern was also observed for items that were designated to be replaced ($t(207) = -2.47$, $P = .014$), compared to baseline items that were not manipulated in that condition. This effect, however, was reversed for suppressed items, which yielded slower RTs relative to baseline ($t(69) = 2.30$, $P = .025$), suggesting that indeed suppressing representations of these items made them more difficult to access. There was no distinction between the two items on clear trials (as both items were to be cleared), so this within-condition comparison was not possible for cleared items. In addition, there were no differences for non-manipulated items across the other three conditions $F(2, 184.18) = 0.126$, $P = .881$), indicating that the baselines against which these effects were calculated did not differ between conditions. Nonetheless, there was a significant difference between manipulated items across conditions ($F(3, 347.98) = 5.211$, $P = .002$), with planned pairwise comparisons demonstrating a significant difference for *maintain* vs. *suppress* ($t(133.94) = -2.30$, $P = .023$) and *suppress* vs. *clear* ($t(122.53) = 2.01$, $P = .047$), but no difference for *maintain* vs. *replace* ($t(129.16) = -0.85$, $P = .40$). Additionally, there were significant interactions between operation and probe type (manipulated, non-manipulated) on RTs for *suppress* vs. *maintain* ($P < .001$) and *suppress* vs. *replace* ($P = .001$). Hence, maintaining an item in WM or having it targeted for replacement with another item allowed participants to more quickly endorse having encoded the item on the current trial. Importantly, only suppressing an item slowed the recognition judgment for that item, and this suggests that suppression produces the most effective removal of information from WM. Together with the neural results showing a selective reduction in proactive interference from suppression (**Fig. 5**), the selective slowing of recognition for suppressed items demonstrates the precise impact which suppression has on the contents of WM. This contrasts with more broad consequences from the suppression of episodic memory *retrieval*, which inhibits hippocampal processing and can produce forgetting of unrelated experiences from around the time of suppression (Hulbert et al., 2016). Understanding the relationship between these unique forms of memory suppression is an exciting area of ongoing research.

[redacted]

Fig 6. Behavioral study procedure and results. **A)** Each trial started with two items (one face and one scene) for 2.76 s, followed by a cue for 1.76 s indicating which cognitive operation should be applied to the item that had appeared at the cued location (i.e., the *manipulated* item). Each cue was linked to one of the four operations based on its shape and color. Then, a 1-s retention interval appeared with central fixation until a single WM probe appeared. The participants were instructed to respond whether they saw the probe item at the beginning of the trial (regardless of whether or not an operation was performed on the item) as fast as possible within 1 s. **B)** Response times (RTs) for correct “yes” probes for different operations by probe type (manipulated, non-manipulated) of the WM items. Data for manipulated items are color-coded by operation and data for non-manipulated items are shown in gray across all operations. *P < .05.

(P. 33 – Supplementary Materials)

Behavioral experiment

Participants

A total of 259 participants (166 female; age, M = 19.25, SD = 2.15) took part in the experiment in exchange for psychology course credit at the University of Colorado Boulder. One participant was excluded due to equipment failure, and 50 participants were excluded due to poor

task performance (< 75% accuracy for positive and/or negative probes), resulting in a final sample of 208 participants (70 Suppress, 67 Clear, 71 Maintain, 208 Replace; 135 female; age, $M = 19.17$, $SD = 1.34$). All participants had normal or corrected-to-normal vision, provided informed consent, and reported no history of brain injury, neurological or psychiatric disorder, nor severe cognitive or psychological problems. The study was approved by the University of Colorado Boulder Institutional Review Board (IRB protocol #18-0571).

Stimuli

Stimuli consisted of colored images of familiar faces (e.g. Ellen DeGeneres) and scenes (e.g. Golden Gate Bridge) that were obtained from various resources including Google Images. There were 252 unique items per each category (504 images total). Importantly none of the images was ever repeated across trials. To prevent stimulus-specific effect (e.g., familiarity), the images were fully randomized across trials, conditions, and participants. For *replace* trials, a subset of the images was used as replacement items (i.e., 24 items per category) and this set was consistent across participants.

Behavioral design and procedure

The study design consisted of a mixed within- and between-subjects design to measure the effects of the different WM operation conditions with unique items across trials. Participants were assigned to one of three groups that differed in the WM operation required: *maintain*, *suppress*, or *clear*. All participants additionally completed trials with the *replace* operation. Prior to the main task, participant completed a familiarization task for the replacement items to ensure that the items could be retrieved based on their names. In this familiarization task, on each trial, a single image appeared at the center of the screen along with two item-name options below the image for 4 s. Participants responded to the name that matched the image using their left or right index finger.

In the main task, participants in each group performed two blocks of trials, one requiring their group-specific operation (e.g., *maintain*) and the other requiring the replace operation. Order of operations was counterbalanced across participants (i.e., half of participants completed the *replace* condition first). Participants received practice trials prior to each block. Each operation consisted of 72 trials for a total of 144 trials, and the task lasted approximately 30 m. On each trial (see **Fig. 6A**), two items (i.e., one face and one scene) were presented for 2.76 s, one to each side of a central fixation cross with position counterbalanced, followed by an instruction screen for 1.76 s with an operation cue on one side of the central fixation. The location of the cue indicated which item had to be manipulated with the given operation. This item is the *manipulated* item while the un-cued item is the *non-manipulated* item. Cues for the *clear* operation appeared in the middle of the screen as both items were to be cleared from WM. The cues appeared in different colors and shapes depending on operations as following: red O for *maintain*, orange <item name of the replacement item> for *replace*, green X for *suppress*, and blue cloud for *clear*. Position of the cue for the *maintain*, *replace* and *suppress* conditions varied randomly between the right and left. A 1-s retention interval followed with a central fixation cross. Afterwards, a probe picture was presented at the center of the screen for 1 s. Participants had to indicate, within 2.5 s, whether the probe had appeared at the beginning of the trial (regardless of whether the item was manipulated or not) with their right or left index fingers, with response mappings counterbalanced across participants. The next trial began after a 1.5 s blank inter-trial interval. Valid probes appeared on 50% of the trials, which consisted of either a

manipulated or non-manipulated item. Negative probes consisted of 66.6% items seen during the previous trial and 33.3% novel items.

Behavioral data analyses

To measure the impact of each operation on the items being manipulated, reaction times (RTs) in the probe task were calculated for correct trials only. RTs below 200 ms or greater than 2.5 standard deviations above the within-subject mean RT were excluded ($M = 1.49\%$ of trials, $SEM = 0.09\%$). First, statistical tests were conducted on the individual mean RT values for correct “yes” response trials. Paired-sample T-tests were applied to compare recognition RTs for manipulated vs. non-manipulated items for each operation (**Fig. 6B**). One-way ANOVAs with *operation* as a within-subjects factor were conducted on RTs separately for manipulated and non-manipulated items. The results were visualized with the `YaRrr` package in R.

Minor Concerns:

1. On p. 8 the authors speculate “Somewhat paradoxically the trajectory for suppress is quite similar to that for maintain, suggesting that a representation must be within the focus of attention in order for it to be suppressed.” The conclusion does not necessarily follow from the data. Moreover, if it did, I see no way in which it makes mechanistic sense. Note that the authors are interpreting BOLD activity which may or may not monotonically relate to neural activity. Active neural firing is not the only metabolically demanding phenomenon, and it is possible that the mechanisms underlying suppression may reduce neural firing, yet increase metabolic demand relative to some resting baseline. Therefore, I would encourage the authors to be a bit more agnostic in their interpretation giving the theoretical peculiarity of positing that a suppressed item is in the focus of attention.

We appreciate the reviewer’s concern, and we have softened our interpretation of the *suppress* results accordingly. It is indeed possible that BOLD signals may not relate monotonically to neural activity. Furthermore, univariate analysis of the BOLD data here showed less activation for *suppress* trials relative to *maintain* trials in the ventral visual stream. This result indicates an overall reduction in neural processing during suppression, and this may be more intuitive and mechanistically plausible. However, making an interpretation based on the univariate results alone is incomplete. When we applied multivariate analyses to these data, we found similar evidence for content-specific information in *maintain* and *suppress* trials. This was true at the category-level (with pattern classification) and the item level (with representational similarity analysis). These multi-variate results, although perhaps counterintuitive, may suggest that suppression promotes “sharpening” of the target representation (Kok, Jehee, & de Lange, 2012) during the process of selective suppression. To effectively suppress the correct representation, it may need to be attended to. This would be consistent with our recent work which found a sharpening of multivariate representations of picture memories during intentional forgetting (Wang, Placek, & Lewis-Peacock, 2019).

(P. 8) Note that we decoded the item-specific neural patterns being represented rather than neural activation intensity per se. Interestingly, the univariate neural activation in the ventral stream was decreased for *suppress* compare to *maintain*, while the multivariate results were equivalent for these two operations. This pattern suggests that suppression may promote “sharpening” of the representation to selectively suppress the target. Repeated presentations of a stimulus, which produces a reduction of activation as assessed by univariate approaches, has been shown to be associated with either increased multivoxel pattern classifier evidence (i.e., representational sharpening), or with decreased evidence (Davis et al., 2014; Kok et al., 2012). Critically, our results propose that suppression may actively identify and target the representation of the item in WM that is to be removed rather than simply inhibiting WM activity in general. This finding is consistent with our recent study demonstrating that intentional forgetting of a picture stimulus produced stronger (“sharper”) multivariate representations of the targeted item during the forgetting attempt (Wang, Placek, & Lewis-Peacock, 2019).

2. On p. 9, the authors interpret differences in the first moment at which removal at the item-level is distinct from maintenance. Given that there are not significant differences among the removal conditions until far later, I do not believe interpreting differences among the conditions is warranted at the item-level. The differences in timing at the category-level appear much clearer and would appear to be directly related to the overall strength of the removal process.

We respectfully disagree with the reviewer on this point. We believe it is an open and interesting question as to whether category- or item-level neural decoding is more closely related to the removal process. Therefore, we would prefer to leave intact Fig. 4B where the decoding time courses are shown at the category- and item-level for the three removal operations. In fact, the differences in decoding *suppress* vs. *maintain* emerge earlier in the category-level results (time widow #1) than in the item-level results (time widow #2). We feel that the qualification of these results, reproduced below, are justified for inclusion.

(P. 9) The discrepancies between the item- vs. category-level analyses regarding onset of removal from attentional focus might be due to reduced sensitivity in these two types of analyses. However, they might also reflect a meaningful cognitive difference such that suppression first impacts item-level details of a stimulus before impacting its general category-level information. Further research is needed to differentiate between these two possibilities.

3. In Fig 4A, the authors show the time course of the “new item” acting as a replacement in the replace condition. It would be useful to perform a surrogate analysis for the other removal conditions as a reference. That is, suppose that the authors randomly assigned a replacement category for the suppress and clear conditions. What would the time course of that surrogate look like? Given that evidence against one category is evidence for other categories, it would be appropriate to perform this analysis to demonstrate that the evidence for the new item in the

replace condition really reflects the new item, and not simply the drop off of evidence for the old item.

We appreciate the reviewer to suggest the surrogate analysis to provide a reasonable baseline. We have adopted this idea and included the baseline time course in the manuscript and in a revised **Fig 4A**.

(P. 8) Furthermore, an increase in classifier evidence was observed for the category of the new item to which individuals switched their attention on *replace* trials. To statistically evaluate this increase, we compared it with an empirical baseline of classifier evidence for an irrelevant category from *suppress* and *clear* trials. This baseline was computed separately for each participant by randomly sampling an irrelevant stimulus category from each *suppress* and *clear* trial (there were no differences between these trial types, $P_s > 0.390$ for all time windows). For example, if Anne Hathaway (a face) was suppressed, we would select randomly the fruit or scene category evidence from that trial to contribute to the baseline. On *replace* trials, the increase in classifier evidence for the replacement item rose significantly above this baseline (windows 4-5, $t_s(49) > 6.73$, $P_s < .001$). This verifies that evidence of “replacement” of one item with another was found only for *replace* trials, and not for either *suppress* or *clear* trials.

(P. 10, Fig 4A) Group-averaged category-level fMRI pattern classifier evidence for the WM item for 14 sec after the onset of each trial, shown separately for the item on maintain trials (red), the item being replaced on *replace* trials (dark orange), the item that serves as a replacement on *replace* trials (light orange), and an empirical baseline of trial-irrelevant items selected randomly from *suppress* and *clear* trials (gray).

4. In the abstract, the authors suggest that the only method to demonstrate that information is removed from mind is self-report. That is a bit overstated. For example, the work by Oberauer shows that lingering representations can be tracked using multiple methods such as the effect of those items on scanning (i.e. load effect) and slowed rejection of those items on probes (i.e. Monsell proactive interference effect). The authors might draw upon such methods for the behavioral study that I recommended. What is true is that the neural mechanisms remain poorly specified, as well as the mapping of neural mechanism to behavior, both of which are novel and impactful enough in themselves.

We appreciate the reviewer to point out these limitations. We have updated the abstract to soften the claim on self-report. As the reviewer suggested, we have also conducted a separate experiment to measure the behavioral impact of these removal processes. Please refer to our response to major concern #1.

(Abstract) Until now, beyond self-report and indirect behavioral measurements, verifying that thoughts have been expunged has been an intractable issue.

5. One matter that would be important to address with behavioral work is whether suppression is beneficial to future memory or detrimental. If it reduces proactive interference, that would suggest a beneficial effect. However, work by Anderson and colleagues (Hulbert et al., 2016) suggests a penumbra effect of suppression information from mind. It would good to reconcile these positions.

As the scope of this study is on working memory removal processes, we didn't hypothesize any effects on long-term memory as investigated by Hulbert et al., (2016). This question remains an open an active area of research. While Hulbert et al. (2016) showed a suppression penumbra effect on episodic memories (i.e., amnesic shadow), there were no reported short-term effects. In our new behavioral study, we found no evidence that suppression of working memory spread to nearby items in WM. We now address these findings and the discrepancy with the Hulbert study.

(P. 16) Together with the neural results showing a selective reduction in proactive interference from suppression (**Fig. 5**), the selective slowing of recognition for suppressed items demonstrates the precise impact that suppression has on the contents of WM. This contrasts with more broad consequences from the suppression of episodic memory *retrieval*, which can produce forgetting of unrelated experiences in close temporal proximity to the time of suppression (Hulbert et al., 2016). Understanding the relationship between these unique forms of memory suppression is an exciting area of ongoing research.

Reviewer #2:

We are pleased that the reviewer found our study to be “impressive in many ways” and we are grateful for the many constructive comments to improve this paper. We take up each point in turn below.

1. First, how do we know the manipulation is working as we would expect? There is no behavioral evidence and the differences between conditions in item decoding wouldn't be expected a priori. I.e., the magnitude of the item removal effect being greater in replace than clear than suppress could indicate a failure of the instructional manipulation. Was any behavioral data collected in this regard?

We agree with the reviewer that the neural differences across operations may not be verified as a removal process without behavioral evidence. Reviewer #1 raised this same concern (see our response to reviewer #1 point #1). As our experiment was designed to maximize our ability to track neural representational changes, we repeated stimuli across all operations and also multiple times in the localizer at the expense of being able to assess trial-specific behavioral consequences. Therefore, to demonstrate that these operations do indeed have behavioral consequences, we conducted a separate behavioral study, which is described in detail in the Supplementary Material, and the main results of which are now presented in a new **Fig. 6**.

The instructions should also be included.

The instructions for the different operations were given on the screen with two operation words in the top and bottom halves of the screen (i.e., maintain, suppress ...) except the *replace* condition in which the word in the bottom half indicated the subcategory of image that the participant should switch to (e.g., “apple”). The participants completed practice trials before the central study and they were informed to retrieve the replacement item that had been shown in the previous localizer (5 times across runs). The details about presenting instructions can be found in the Materials and Methods section of the Supplementary Materials (P. 23).

2. While the WM operation decoding analysis suggests that people were indeed performing qualitatively different tasks in the 4 conditions rather than e.g., different degrees of something similar, there is still the possibility that this decoding is picking up on perceptual differences of the cue words. Some effort could be made to remove the cue period from the decoding analysis and/or remove perceptually sensitive regions from the classifier. Another approach might be to test the classifier across participants, which should presumably be less sensitive to visual word differences between conditions. This could also help validate the instructional manipulation.

We agree that it is possible for the WM operation classifiers to leverage other information in the BOLD signal that may not be due to cognitive operations per se. For example, the operation cues varied visually and semantically across operations, and these perceptual differences could be driving classification of the operations. We have followed the reviewer's suggestion and tested the operation classifiers across participants. This was done both using anatomical alignment across participants, and also functional alignment (using the "hyperalignment" procedure from Haxby et al., 2011). These approaches yielded similar results, and thus we report the simpler version with anatomical alignment. Across-participant classification of operations was indeed successful for each operation (shown now in **Fig. 2A**). These results dovetail with our already reported data showing systematic similarities, throughout the brain, across participants in multivariate classifier importance maps (**Fig. 2B**) and univariate activation responses for the different cognitive operations (**Fig. S1**). We believe these data provide convincing evidence that the WM operation classifiers are not merely picking up on perceptual differences of the cue words.

(P. 4) We replicated these within-subject classification results using across-subject classifiers (accuracy $M = 0.379$, $SD = 0.086$, all $P_s < .001$, **Fig. 2A**, right) in which all individuals' voxels in MNI space were aligned anatomically. The success of this procedure, combined with the group-level univariate results and classifier importance maps, demonstrates that similar neural processes were recruited for each operation across participants.

The procedures for this analysis are now described in the Methods section:

(P. 27) Across-subject WM operation classification

The BrainIAK toolbox (Kumar et al., 2020) for Python was used for across-subject pattern classification analyses with L2-regularized (penalty = 50), non-multinomial (one-vs-others, for each category) logistic regression. All data from all participants ($N=49$) were normalized to MNI standard brain space and concatenated, so that all voxels are anatomically aligned across participants. The first half of all runs was used for feature-selection, and the other half of the data was used for training and testing the classifier. One participant was excluded from this analysis due to missing half of their data. The top 10,000 voxels were first feature-selected from the whole-brain gray mask segmented from standardized MNI brain (17.13% of 58,229 voxels, ANOVA threshold: $P < .001$), and the feature dimensions were then reduced to 70 components using principal component analysis (PCA) provided in BrainIAK. This number of components was selected as the optimal value to maximize classification accuracy using a repeated cross-validation testing scheme. The same feature-selected voxels of the testing data were transformed to component space and used for 49-fold leave-one-participant-out cross-validation across all participants, with the operation regressor shifted forward 4.6 s to account for hemodynamic lag.

We also conducted another version of across-subject classification with functionally aligned data using the hyperalignment procedure (Haxby et al., 2011) in BrainIAK. The first half of the data was used for feature selection (top 5,000 voxels per hemisphere in each individual's native brain space, 26.71% of $M = 37,436$ voxels, ANOVA threshold: $P = .005$) and hyperalignment, and the other half was used for classification. To create the template (i.e., common space) for each hemisphere to which all participants' voxels were aligned, we performed two steps. First, each individual's voxels were aligned to a "reference," which was the first participant with a full dataset, using Procrustean transformation. Then, for each step in this iterative procedure, the reference was updated by averaging the current reference with the newly aligned participant. In the second step, each individual's voxels were aligned again to the reference that was obtained from the completion of the first step. Then, all individual's features in the common space were averaged, and this group-averaged feature set served as the final reference template. Finally, we obtained transformation parameters for each participant, using this final template, and then transformed the feature-selected voxels of testing data into the hyperaligned common space. The hyperaligned features were then reduced using PCA (70 components, which was also the optimal value) and then used for cross-validation classification. The classifier accuracy was reliably above chance for both of the anatomically aligned ($M = 0.379$, $SD = .086$, one-sample t test: omnibus $T_s(48) > 6.49$, $P_s < .001$) and hyperaligned across-subject classifiers ($M = 0.378$, $SD = .097$, $T_s(48) > 5.35$, $P_s < .001$) across all operations (**Fig 2A**).

3. The subsequent proactive interference encoding fidelity analysis is a nice idea, but there are conceptual and interpretational concerns. Conceptually, while similarity could lead to interference, it is unclear whether category-level similarity is sufficient to cause interference here or whether there could even be the opposite -- a congruency benefit. Indeed, the authors argue that category-level congruency could be driving the post-suppression item representational enhancement effect.

We understand that our results may seem paradoxical as category-level congruency could cause either an interference or enhancement effect (e.g., priming). While it has been suggested that overlapping features between related items would compete for WM capacity (Cohen, Konkle, Rhee, Nakayama, & Alvarez, 2014), it is unclear how the suppression of different levels of features (item-specific vs. category-general) affects subsequent encoding in WM. Regardless, our data differentiate suppression from the other removal operations evaluated here. Understanding the precise mechanism underlying the WM suppression operation is a valuable aim, but is beyond the scope of this study, and is left for the focus of a future investigation.

4. Regarding the interpretation, the lack of a behavioral measure of success makes this quite challenging given that the neural results are not clearly consistent. The suppress condition shows the least evidence for removal in the decoding analyses, but then the least evidence of proactive

interference on fidelity. These results are difficult to reconcile. One possibility is that these effects are driven by differences in difficulty, where “suppress” is the most difficult operation. That would explain the decoding effects and could potentially account for the interference analysis as well (see next comment). Is there any way to rule out such an explanation?

Our new behavioral results show that suppressed items are recognized more slowly than baseline (non-manipulated) items in a standard test of WM recognition test. This result is consistent with the idea that the suppressed item was actively inhibited during the removal operation. The paradoxical but interesting result from the fMRI data was that the suppressed item was actively represented (rather than inhibited) during its removal, and this led to better neural encoding on the next trial. This finding is consistent with our recent evidence using an item-method direct-forgetting paradigm which showed that to-be-forgotten items had more active representation (based on fMRI pattern classification), and yet worse subsequent memory, compared with to-be-remembered items (Wang et al., 2019). The neural enhancement observed with suppression in these couple of studies is a focus of our ongoing research.

(P. 8) This is consistent with our recent study which demonstrated that intentional forgetting of a picture stimulus produced stronger (“sharper”) multivariate representations of the targeted item during the forgetting attempt (Wang et al., 2019).

5. In figure 5b, same and different should be plotted separately. This is critical, because it is possible that differences in “fidelity” of the different categories is driving the effect of operation, which could undermine the argument that “suppress” is the most effective operation. For example, if “suppress” is the most difficult operation, it might interfere with the subsequent trial for different categories, which could enhance the same - different value. Of course, there should still be a control for overall similarity in the fidelity measure, but this is better captured by subtracting out similarity with other (related) non-target items.

We appreciate the reviewer pointing out that the statistics for the baseline (i.e., different category) was not reported. There were no reliable differences for the different-category trials across operations and the results remained the same when comparing only same-category trials. We reported *same-minus-different* results to show facilitation vs. interference effect in more efficient way (Fig. 5B). We are glad that the reviewer highlighted this issue and now the baseline statistics are included in the manuscript.

(P. 14) Note that there were no reliable differences for the different-category encoding fidelity across operations indicating an equivalent baseline ($P_s > .096$), with differences only observed for the same-category analysis ($P_s < .001$).

6. The authors should examine whether reduced fidelity in the proactive interference analysis could be driven by category-level repetition suppression effects. That is, there may be a reduced overall neural response to an item from a category or subcategory that just appeared. Comparing the effects of reduced univariate activation to reduced item representational fidelity would inform the nature of the suppression.

If the subsequent encoding effects (i.e., representational fidelity on N+1 trials) came from category-level repetition suppression, we would expect both *suppress* and *maintain* conditions to have large drops in representational fidelity. This prediction comes from the category-level decoding results in Fig 4B, where these two conditions have similar degrees of category decoding during the operation. However, our results show only a selective drop in fidelity for suppression. Moreover, we quantified the encoding fidelity with an item-level RSA analysis that was weighted with item-specific betas. This analysis was therefore sensitive to unique activity patterns for each item rather than generic category-level features. (Note that items within subcategory are distinct from each other, Fig. 3B). We have clarified this issue in the manuscript to rule out the repetition suppression account

(P. 8) Note that we decoded the item-specific neural patterns being represented rather than neural activation intensity per se. Interestingly, the univariate neural activation in the ventral stream was decreased for *suppress* compare to *maintain*, while the multivariate results were equivalent for these two operations. This pattern suggests that suppression may promote “sharpening” of the representation to selectively suppress the target. Repeated presentations of a stimulus, which produces a reduction of activation as assessed by univariate approaches, has been shown to be associated with either increased multivoxel pattern classifier evidence (i.e., representational sharpening), or with decreased evidence (Davis et al., 2014; Kok et al., 2012). Critically, our results propose that suppression may actively identify and target the representation of the item in WM that is to be removed rather than simply inhibiting WM activity in general. This finding is consistent with our recent study demonstrating that intentional forgetting of a picture stimulus produced stronger (“sharper”) multivariate representations of the targeted item during the forgetting attempt (Wang, Placek, & Lewis-Peacock, 2019).

7. The authors state that because “replace category” was indistinguishable from “replace subcategory,” they removed the latter from the main results. I don’t understand the logic here – why not collapse these two conditions?

We understand this confusion and agree that we should make the logic clearer. We didn’t collapse the two replace conditions because we did not have the analysis sensitivity to properly decode the “replace subcategory” trials. We ended up focusing on category-level decoding, rather than subcategory-level decoding (which proved to be insufficiently powered), and thus we were unable to separately decode the old and new items on “replace subcategory” trials. Also,

collapsing the two conditions would have resulted in unbalanced trial numbers across operations, which would bias the operation classifiers towards this class. We clarified this logic in the manuscript.

(P. 26) We focused on category-level neural decoding (subcategory-level decoding was insufficiently powered) and *replace subcategory* trials were not suited for this analysis. For this reason, and to avoid biasing the operation classifiers, we excluded *replace subcategory* data rather than combining it with *replace category* data.

8. It's unclear how the optimal penalty was chosen for each participant. The description makes it sound as though the penalty with the best cross-validation performance was chosen but that would inflate cross validation accuracy (i.e., double dipping). Was there a separate held out set? In addition, it's unclear how this procedure would result in a single penalty per subject.

All classifiers were verified with k-fold leave-one-out cross validation in which the classifier was trained with $k-1$ runs and tested on the held-out run for each iteration to prevent double dipping. The optimal penalty value was obtained based on the classifier accuracy from testing on the held-out sets. A single penalty was chosen for each subject based on the maximum generalization performance from this iterative penalty search. Nonetheless, to address any concern with double dipping, we replicated our analyses for both the WM operation classifiers and the WM representation classifiers with a single penalty value of 50 for all participants, and the results did not change.

(P. 27) To find the optimal L2 penalty value for each classifier's best fitting model, the cross-validation was done with different penalties in two steps: (1) eight iterations with a broad range of penalties (from 0 to 10,000 with exponential increase) and then (2) 10 iterations in a narrow range around the best penalty value from the first step. A single penalty was chosen for each subject based on the maximum generalization performance from this iterative penalty search.

(P. 28) Additionally, the WM representation classifier was replicated with a single penalty value of 50 across participants, and the classifier accuracy and sensitivity remained reliably above chance at the category level (accuracy: $M = 0.80$, $SEM = 0.014$, $P_s < .001$; AUC: $M = 0.91$, $SEM = 0.009$, $P_s < .001$) and at the subcategory level (accuracy: $M = 0.32$, $SEM = 0.011$, $P_s < .001$; AUC: $M = 0.75$, $SEM = 0.010$, $P_s < .001$).

9. The authors should include non-baseline corrected timecourses in supplement – why should there be any difference from 0 on average?

The neural signals at the onset of a trial can vary slightly due to gradual drifting of signal intensity across time, or from lingering signals from the previous trial. We performed baseline normalization to compensate for this. However, we also show here the uncorrected version. The results are so similar to the corrected version reported in the main text that we did not feel it warranted to put this into the supplement, but will be happy to do so if the reviewer thinks this would be beneficial. We have also addressed this issue in the text.

[redacted]

Fig R4A (uncorrected version of **Fig. 4A**). Group-averaged category-level fMRI pattern classifier evidence for the WM item for 14 s after the onset of each trial for the four cognitive operations. Data were not baseline corrected at the beginning of the trial. All results reported in the main analysis hold.

(P. 28) The data were baseline corrected by removing the mean target classifier evidence, separately for each condition, from the first 2.76 s (6 TRs) from all subsequent time points. This procedure had no effect on any statistical comparisons between conditions but centered the data at trial onset to 0 rather than 0.45 (classifier evidence, ranging from 0 to 1).

10. The regions that were used for each analysis should be included in the figures.

We used the ventral visual stream (VVS) ROI for the WM representation decoding in category (i.e., classifier) and item (i.e., RSA) levels and the whole-brain ROI with gray matter for the WM operation decoding. These ROIs were included in **Fig. 1B**. We have clarified the ROIs in the figure caption for **Fig. 1** and **Fig. 3**.

(Fig. 1B) Note that whole brain data was used for the subcategory classifier to capture semantic differences across subcategories (e.g., actors vs. musicians).

(Fig. 3) **A)** Classifier confusion matrices for category-level classification (in the ventral visual stream, VVS) and subcategory-level classification (whole brain), **B)** and item-level RSA (VVS) on the localizer data.

11. Lastly, because there are no behavioral outcome measures here, the level of discussion of potential therapeutic implications seems inappropriate.

With the inclusion of our new behavioral data, we believe that this discussion is now more appropriate to include.

Additional updates

1. We originally used the whole-brain mask including gray and white matters, which we believed to be excluded via feature-selection. To report more accurate classifier performance, we updated the whole-brain mask only including gray matter (segmented in individual's subject space). The results remained the same but the details, such as digit numbers in statistics and number of selected voxel numbers, were slightly changed.

(P. 27) To construct this mask for each participant, thresholded masks of these regions (bilaterally) were summed together and converted to individual's native brain space. The whole-brain mask consists of only gray matter that was segmented based on a high-resolution structural brain image using FMRIB's automated segmentation tool (FAST) provided in FSL. The ROI masks were then binarized so that voxels within the mask had a value of 1 and voxels outside of the mask had a value of 0 (VVS: $M = 13,825$, $SD = 1,514$ voxels; whole-brain: $M = 37,343$, $SD = 2,962$, **Fig. 1B**).

2. We removed the subset of volumes ($M = 11.44$ TRs, $SD = 17.81$) due to excessive head motion. The results remain the same except the specific numbers have slightly changed, and have been updated throughout the text.

3. We updated a few parts that were incorrect as follows.

Fig. 1: item -> encoding

Fig 4. Bonferroni > FDR

4. We included image processing tools and software packages for generating figures in the supplementary to meet image integrity policy.

- (P. 31) ... and visualized using FreeSurfer (**Fig. 2B**)
- (P. 33) ...we applied representational similarity analysis (RSA) with custom code in MATLAB
- (P. 33) ... we modeled beta estimates using SPM12
- (P. 36) The results were visualized with the YaRrr package in R.

References

- Cohen, M. A., Konkle, T., Rhee, J. Y., Nakayama, K., & Alvarez, G. A. (2014). Processing multiple visual objects is limited by overlap in neural channels. *Proceedings of the National Academy of Sciences of the United States of America*, *111*(24), 8955–8960. <https://doi.org/10.1073/pnas.1317860111>
- Davis, T., LaRocque, K. F., Mumford, J. A., Norman, K. A., Wagner, A. D., & Poldrack, R. A. (2014). What do differences between multi-voxel and univariate analysis mean? How subject-, voxel-, and trial-level variance impact fMRI analysis. *NeuroImage*, *97*, 271–283. <https://doi.org/10.1016/j.neuroimage.2014.04.037>
- Haxby, J. V., Guntupalli, J. S., Connolly, A. C., Halchenko, Y. O., Conroy, B. R., Gobbini, M. I., ... Ramadge, P. J. (2011). A common, high-dimensional model of the representational space in human ventral temporal cortex. *Neuron*, *72*(2), 404–416. <https://doi.org/10.1016/j.neuron.2011.08.026>
- Kok, P., Jehee, J. F. M., & de Lange, F. P. (2012). Less is more : expectation sharpens representations in the primary visual cortex. *Neuron*, *75*, 265–270. <https://doi.org/10.1016/j.neuron.2012.04.034>
- Kumar, M., Ellis, C. T., Lu, Q., Zhang, H., Capotã, M., Willke, T. L., ... Norman, K. A. (2020). BrainIAK tutorials: User-friendly learning materials for advanced fMRI analysis. *PLoS Computational Biology*, *16*(1), e1007549. <https://doi.org/10.1371/journal.pcbi.1007549>
- Wang, T., Placek, K., & Lewis-Peacock, J. (2019). More is less: increased processing of unwanted memories facilitates forgetting. *Journal of Neuroscience*, *39*(18), 3551–3560. <https://doi.org/https://doi.org/10.1523/JNEUROSCI.2033-18.2019>

Reviewers' Comments:

Reviewer #1:

Remarks to the Author:

In my previous review, I raised the concern that behavioral data were needed to bolster the excellent neuroimaging findings regarding the different mechanisms of removing information from mind. The authors were responsive to my concern, as well as my other more minor concerns. Overall, I remain highly enthusiastic about this study, and I appreciate the authors' efforts – especially the collection of such a large behavioral dataset. Unfortunately, I am not certain how to interpret the new behavioral study. As I understand it, participants were asked to remove information from working memory. Thereafter, they were asked to respond positively to items they were told to remove from working memory. These two demands seem to clash – to successfully perform the latter, they must not successfully perform the former. Therefore, I am not sure what behavioral effect the authors are measuring here and whether/how it relates to proactive interference. The increased RT to suppressed items could reflect conflict regarding whether to respond to suppressed items positively or negatively, or potentially a demand characteristic.

Given the experimental details, my guess is that the authors had planned to examine recent negatives, which is often used to study proactive interference behaviorally. Perhaps nothing came of that analysis, which could potentially be due to confusion on the part of the participants of whether or not to truly remove information from working memory. One way around that confusion would be to not probe the replaced and suppressed items on the trial itself (i.e. only non-manipulated items are positive probes), but allow those items to be lures in the following trial (i.e. requiring a negative response). Clear trials could be probed with a memory neutral item (e.g. left or right arrow), but similarly could be lures on the following trial. If suppression uniquely reduces proactive interference as the authors surmise, then the recent negatives interference effect would be predicted to be reduced for previously suppressed items relative to previously replaced and/or cleared items.

Alternatively, the authors could test the removed items as lures on the same trial that they are removed (e.g. Nee et al., 2007, *NeuroImage*; Nee & Jonides, 2008, *Psych Sci*). In that case, the appropriate response to suppressed/replaced items would be negative, which typically results in a slowed negative response relative to novel items. A reduction in that slowing would indicate superior removal. However, I am uncertain how to include a suitable clear condition for this kind of a setup.

Another possibility that follows closely from the neuroimaging data could test for release from proactive interference following suppression by examining performance on same vs. different category items. Either way, I think more convincing behavioral data are needed here.

Given the current situation with COVID-19 and the inability of the authors to collect new data at the present, I would also be open to hearing a rebuttal from the authors regarding why the present behavioral paradigm is suitable in light of my concerns. In that case, I would also like to see error-rate data and inverse efficiency scores to ensure that the results are not driven by speed-accuracy tradeoffs. I think that is particularly important given the between-subject nature of the design and the possibility that some of the instructions may have caused confusion for the participants.

Reviewer #2:

Remarks to the Author:

The authors have addressed all my major concerns. I agree the addition of the figure of non-baseline corrected data is unnecessary as there are no concerning baseline differences. The description in the

text is sufficient.

The additional behavioral study really strengthens the paper. In particular, the suppress effect showing a significant slowing in recognition is key. Just a couple suggestions:

One additional analysis that would help strengthen the paper would be to test whether there is a behavioral advantage that mirrors the advantage of suppress on neural proactive interference (Fig 5b). That is, does suppressing a face on trial N improve lead to faster recognition of a face on trial N+1 compared to other operations?

The statement that because the suppress condition is the only one to show significant slowing, this "suggests that suppression produces the most effective removal of information from WM" is not quite supported by the behavioral data alone. That is, couldn't the faster recognition performance on replace trials be supported by retrieval from long-term memory, perhaps facilitated by the association with the new cue? If replace-cued items show proactive interference into the next trial compared to suppress-cued items, that would support the idea that the replace items are still in working memory.

Other suggestions:

Follow-up to R1's point – the line of the abstract "Until now, beyond self-report and indirect behavioral measurements, verifying that thoughts have been expunged has been an intractable issue." I would qualify "expunged" with "neurally" or "in the brain," Otherwise it sounds like you are suggesting that fMRI is a more direct measure of "thought" than behavioral probes, which would be a pretty tough argument to make.

The addition of the empirical baseline to compare with the replacement category in Fig 4a is nice. It would be great to add the statistics of that effect to the figure.

Reviewer #1 (Remarks to the Author):

In my previous review, I raised the concern that behavioral data were needed to bolster the excellent neuroimaging findings regarding the different mechanisms of removing information from mind. The authors were responsive to my concern, as well as my other more minor concerns. Overall, I remain highly enthusiastic about this study, and I appreciate the authors' efforts – especially the collection of such a large behavioral dataset. Unfortunately, I am not certain how to interpret the new behavioral study. As I understand it, participants were asked to remove information from working memory. Thereafter, they were asked to respond positively to items they were told to remove from working memory. These two demands seem to clash – to successfully perform the latter, they must not successfully perform the former. Therefore, I am not sure what behavioral effect the authors are measuring here and whether/how it relates to proactive interference.

We thank the reviewer for their enthusiasm for our study and our revision efforts. It is now clear to us that we did not sufficiently justify the intended role of the new behavioral data. These data were meant to complement the neural data, not to replicate them. That is, rather than identifying behavioral correlates of our neural findings (i.e., reduced proactive interference as a result of suppression), our goal for the behavioral study was to assess whether these removal operations yielded any behavioral consequences whatsoever. To amplify this point, in the neuroimaging study we specifically did not want to query an individual's memory of an item on a given trial. We made this design decision because inserting such a query as that would have precluded us from examining how the removal operations influenced encoding on the subsequent trial (i.e., that query would have been an intervening and confounding event). Hence, the behavioral study was designed to examine the effect that we could not examine by design in the neuroimaging study. That is, it was designed to examine how each of the removal operations influences the ability to make a subsequent WM decision about an item that was manipulated. **This motivation has been clarified in the manuscript on p. 16-17.**

The increased RT to suppressed items could reflect conflict regarding whether to respond to suppressed items positively or negatively, or potentially a demand characteristic.

We are confident that misunderstanding of task instructions was not an issue for several reasons: First, we had identified participant confusion regarding instructions as a potential issue when our first round of pilot data with the task yielded some participants who performed well and others who had very low accuracy. In evaluating these initial findings, the pattern of performance on low accuracy trials seems to suggest that the individuals with low accuracy were showing a preponderance of incorrect “No” responses (the correct response was “Yes”) when the probe was the unmanipulated item on that trial. As a result, the instructions were subsequently modified to make explicit that participants should respond “Yes” to items that belonged to each trial *regardless of whether or not the item was manipulated*. This instruction modification resulted in greatly increased accuracy, and we no longer found the pattern of “No” responses to “Yes” trials when the probe was the unmanipulated item. Second, we excluded participants with accuracy lower than 75% on any trial type to ensure that we did not analyze data from participants who may have been confused about the task. **These two points have been added to the manuscript on p. 17.**

With regard to potential demand characteristic issue—if our suppress results were indeed the result of a demand characteristic, we would expect to see a similar result with cleared items because during both operations participants are instructed to remove the information from working memory. Because the RTs of cleared items are not slowed like the suppressed items, we feel confident that demand characteristics are not driving the suppress results.

We agree that possible speed-accuracy tradeoffs are important to consider. However, participants performed exceedingly well on the task, such that no statistically significant differences were noted across conditions. Nonetheless, below are the error rates which show that suppressed items have *lower accuracies* (numerically) in addition to slower response times. Similarly, maintained items have *higher accuracies* (numerically) in addition to faster response times. These are in the opposite direction of a speed-accuracy tradeoff, and as such, we are confident that our results are not due to speed-accuracy tradeoffs.

Given the experimental details, my guess is that the authors had planned to examine recent negatives, which is often used to study proactive interference behaviorally. Perhaps nothing came of that analysis, which could potentially be due to confusion on the part of the participants of whether or not to truly remove information from working memory. One way around that confusion would be to not probe the replaced and suppressed items on the trial itself (i.e. only non-manipulated items are positive probes), but allow those items to be lures in the following trial (i.e. requiring a negative response). Clear trials could be probed with a memory neutral item (e.g. left or right arrow), but similarly could be lures on the following trial. If suppression uniquely reduces proactive interference as they authors surmise, then the recent negatives interference effect would be predicted to be reduced for previously suppressed items relative to previously replaced and/or cleared items.

Alternatively, the authors could test the removed items as lures on the same trial that they are removed (e.g. Nee et al., 2007, NeuroImage; Nee & Jonides, 2008, Psych Sci). In that case, the appropriate response to suppressed/replaced items would be negative, which typically results in a slowed negative response relative to novel items. A reduction in that slowing would indicate superior removal. However, I am uncertain how to include a suitable clear condition for this kind of a setup.

Another possibility that follows closely from the neuroimaging data could test for release from proactive interference following suppression by examining performance on same vs. different category items. Either way, I think more convincing behavioral data are needed here.

As we explain above, and now clarify in the manuscript, our primary goal with the behavioral study was to evaluate direct evidence of behavioral consequences of these removal operations, rather than indirect consequences such as proactive interference. We used a modified Sternberg working memory task to evaluate RTs to Yes/No probes following each of the operations. Indeed, we included a ‘recent negatives’ manipulation to allow for the opportunity to analyze both trial N and N+1 data. However, the trial N+1 analysis yielded null results that are difficult to interpret. This is likely because testing on trial N contaminated the testing on trial N+1 (as discussed in our response above). Thus, in the paper we focused only on the behavioral results from trial N. Note that trial N+1 was not contaminated in the neural study (where we report a proactive interference effect), as there was no test on the preceding trial N.

We appreciate the reviewer’s suggestions for alternate approaches to test for proactive interference. We feel that pursuing these ideas is outside the scope of this paper, but we appreciate these ideas as they are helpful in our planning for follow-up research.

Given the current situation with COVID-19 and the inability of the authors to collect new data at the present, I would also be open to hearing a rebuttal from the authors regarding why the present behavioral paradigm is suitable in light of my concerns. In that case, I would also like to see error-rate data and inverse efficiency scores to ensure that the results are not driven by speed-accuracy tradeoffs. I think that is particularly important given the between-subject nature of the design and the possibility that some of the instructions may have caused confusion for the participants.

We thank the reviewer for their flexibility on this issue. We feel that we have addressed these concerns above, and the resulting updates to the manuscript have helped to clarify that the inclusion of the behavioral data provides data that is complementary to the neural data.

=====

Reviewer #2 (Remarks to the Author):

The authors have addressed all my major concerns. I agree the addition of the figure of non-baseline corrected data is unnecessary as there are no concerning baseline differences. The description in the text is sufficient.

The additional behavioral study really strengthens the paper. In particular, the suppress effect showing a significant slowing in recognition is key. Just a couple suggestions:

We are thankful that the reviewer finds the paper to be strengthened, we agree!

One additional analysis that would help strengthen the paper would be to test whether there is a behavioral advantage that mirrors the advantage of suppress on neural proactive interference (Fig 5b). That is, does suppressing a face on trial N improve lead to faster recognition of a face on trial N+1 compared to other operations?

This is a very reasonable suggestion. As we explain above in response to R1, the purpose of our behavioral study was not to find a behavioral correlate of proactive interference, but rather to test for immediate and direct behavioral consequences of the removal operations.

The statement that because the suppress condition is the only one to show significant slowing, this “suggests that suppression produces the most effective removal of information from WM” is not quite supported by the behavioral data alone. That is, couldn’t the faster recognition performance on replace trials be supported by retrieval from long-term memory, perhaps facilitated by the association with the new cue? If replace-cued items show proactive interference into the next trial compared to suppress-cued items, that would support the idea that the replace items are still in working memory.

We appreciate the reviewer’s point that this statement was too strong based on the behavioral data alone. **We have adjusted this statement accordingly on p. 18:** “Importantly, only suppressing an item slows the recognition judgment for that item. Combined with the neural finding that only suppression reduces proactive interference, this finding suggests that suppression produces the most effective removal of information from WM.”

Other suggestions:

Follow-up to R1’s point – the line of the abstract “Until now, beyond self-report and indirect behavioral measurements, verifying that thoughts have been expunged has been an intractable issue.” I would qualify “expunged” with “neurally” or “in the brain,” Otherwise it sounds like you are suggesting that fMRI is a more direct measure of “thought” than behavioral probes, which would be a pretty tough argument to make.

This is an important clarification. We have added the phrase “in the brain” to the abstract as suggested.

The addition of the empirical baseline to compare with the replacement category in Fig 4a is nice. It would be great to add the statistics of that effect to the figure.

We have updated Fig. 4 with a “triangle” marker to indicate statistical significance and the legend has been updated accordingly: “Triangle indicates the “replace-new start point” defined as the first 3-TR window (1.38 s) in which the decoding evidence for the replace-new item was reliably greater than baseline.”

Reviewers' Comments:

Reviewer #1:

Remarks to the Author:

The authors have made a cogent rebuttal to my concerns. Although I maintain it would be preferable to observe signatures of proactive interference in the behavioral study, the authors have convinced me that the behavioral study as collected appropriately bolsters the neuroimaging findings. Therefore, I am happy to recommend publication of this manuscript. I commend the authors on a job well done on a study that is sure to impact the field.